# Serial millisecond crystallography for routine room-temperature structure determination at synchrotrons

Tobias Weinert[1], Natacha Olieric[1], Robert Cheng[2], Steffen Brünle[3], Daniel James[1], Dmitry Ozerov[4], Dardan Gashi[1,5], Laura Vera[6], May Marsh[6], Kathrin Jaeger[1], Florian Dworkowski[6], Ezequiel Panepucci[6], Shibom Basu[6], Petr Skopintsev[1], Andrew S. Doré[7], Tian Geng[7], Robert M. Cooke[7], Mengning Liang[8], Andrea E. Prota[1], Valerie Panneels[1], Przemyslaw Nogly[1], Ulrich Ermler[3], Gebhard Schertler[1,9], Michael Hennig[2], Michel O. Steinmetz[1,10], Meitian Wang[6] & Jörg Standfuss[1]

Historically, room-temperature structure determination was succeeded by cryo-crystallography to mitigate radiation damage. Here, we demonstrate that serial millisecond crystallography at a synchrotron beamline equipped with high-viscosity injector and high frame-rate detector allows typical crystallographic experiments to be performed at room-temperature. Using a crystal scanning approach, we determine the high-resolution structure of the radiation sensitive molybdenum storage protein, demonstrate soaking of the drug colchicine into tubulin and native sulfur phasing of the human G protein-coupled adenosine receptor. Serial crystallographic data for molecular replacement already converges in 1,000–10,000 diffraction patterns, which we collected in 3 to maximally 82 minutes. Compared with serial data we collected at a free-electron laser, the synchrotron data are of slightly lower resolution, however fewer diffraction patterns are needed for de novo phasing. Overall, the data we collected by room-temperature serial crystallography are of comparable quality to cryo-crystallographic data and can be routinely collected at synchrotrons.

[1] Laboratory of Biomolecular Research, Division of Biology and Chemistry, Paul Scherrer Institut, 5232 Villigen PSI, Switzerland. [2] LeadXpro AG, Park InnovAARE, 5234 Villigen PSI, Switzerland. [3] Molecular Membrane Biology, Max-Planck Institute of Biophysics, Max-von-Laue-Straße 3, 60438 Frankfurt, Germany. [4] Science IT, Paul Scherrer Institut, 5232 Villigen PSI, Switzerland. [5] SwissFEL, Paul Scherrer Institut, 5232 Villigen PSI, Switzerland. [6] Macromolecular Crystallography, Swiss Light Source, Paul Scherrer Institut, 5232 Villigen PSI, Switzerland. [7] Heptares Therapeutics Ltd, Biopark Broadwater Road, Welwyn Garden City AL7 3AX, UK. [8] Linac Coherent Light Source, SLAC National Accelerator Laboratory, 2575 Sand Hill Road, Menlo Park, CA 94025, USA. [9] Department of Biology, ETH Zurich, 8093 Zürich, Switzerland. [10] University of Basel, Biozentrum, Basel 4056, Switzerland. Correspondence and requests for materials should be addressed to J.S. (email: joerg.standfuss@psi.ch)

Serial femtosecond crystallography (SFX) at X-ray free-electron lasers (XFELs) is a unique method[1] for the determination of challenging structures without radiation damage at room-temperature[2]. In particular, the development of high-viscosity injectors that have markedly reduced sample consumption[3] has opened the door for the structure determination of novel membrane proteins[4] and protein complexes[5], and de novo phasing of pharmacological targets[6]. Room-temperature structure determination is of increasing importance in the elucidation of protein dynamics. As opposed to cryo-crystallography, room-temperature X-ray structures reveal realistic conformational flexibility that is crucial for protein function[7, 8]. In addition, room-temperature crystallography enables time-resolved studies[9] that reveal protein dynamics[10] and enzyme catalysis[11] with atomic detail. However, since the beamtime available at XFELs will remain scarce for the foreseeable future, routine room-temperature structure determination at these sources is difficult to achieve. A promising solution is to adapt the methodology developed for macromolecular crystallography at XFELs to synchrotron sources, where radiation damage cannot be outrun but where the radiation dose per crystal can be reduced by using many crystals.

Here, the term serial crystallography is commonly used in a broader sense, covering loop- or sandwich-scanning techniques[12–14] as well as the combination of a rotation series with the use of dozens to hundreds of individual crystals[15, 16]. The most direct approach is serial millisecond crystallography (SMX), which utilizes the same high-viscosity injectors successful at XFELs to distribute the radiation dose over thousands of crystals to determine room-temperature structures with minimum radiation-damage.

In pioneering SMX experiments, data were collected continuously (shutterless), exposing one crystal per image with a $10 \times 30 \, \mu m^2$ beam on a PILATUS 6 M detector with a low frame rate of 10 Hz[17]. During the revision of this manuscript a study performed at advanced photon source was published, describing data collection at 10Hz with a PILATUS3 6 M detector[18]. In another approach, data were collected with a very small $2 \times 3 \, \mu m^2$ beam, using a shutter to allow short exposure times between 10 and 50 ms and a Rayonix detector with a maximal frame rate of 17 Hz. In these approaches data collection times were very long and took up to 5 days in case of the membrane protein bacteriorhodopsin. Here, we demonstrate how the combination of a high-viscosity injector[3] with the high frame rates of an EIGER 16 M detector[19] in a 'crystal scanning' approach allows SMX to be routinely applied to the three most common crystallographic techniques: high-resolution structure determination, ligand soaking and de novo phasing (Fig. 1). To demonstrate the general applicability of SMX, we have chosen three real-life targets: the radiation sensitive molybdenum storage protein (MOSTO), the αβ-tubulin-darpin complex (TD1) soaked with the drug colchicine and the thermostabilised adenosine $A_{2A}$ G protein-coupled receptor ($A_{2A}R$). All individual experiments including native single-wavelength

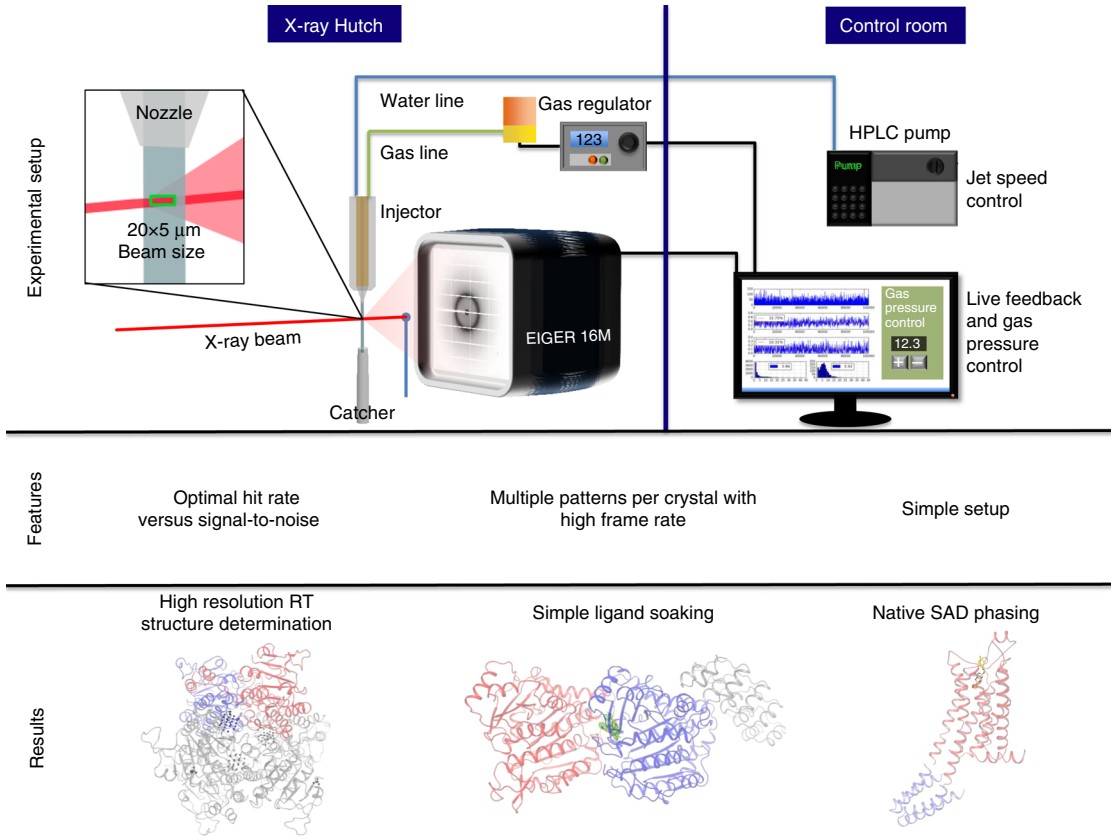

**Fig. 1** SMX setup, features and overview of results. The schematic of the SMX experimental setup at the Swiss Light Source shows the high-viscosity injector set to extrude a 50 μm diameter stream of crystal laden LCP (or other viscous carrier) vertically through the interaction region with the X-ray beam and into a catcher for safe and clean disposal of material. The micro-focused beam is centered on the extruded medium ~50 μm below the nozzle. The jet speed determines exposure time and is calculated from the nozzle diameter and sample flow rate (which is calculated from the pump flow rate). The extrusion is controlled with a helium sheath, which can be adjusted with the pressure regulator to maximize jet stability. A typical beam size in our experiments is 20 μm horizontal and 5 μm vertical with respect to the extruded medium. This size allows for a large interaction volume and maximum flux. Data are sent to the beamline data collection servers where real-time peak finding is fed back to the user

anomalous dispersion (SAD) phasing can be performed in less than a single 8-h synchrotron shift, with data convergence of native data sets already being achieved after maximally 82 min.

## Results

**Optimizing data collection parameters and radiation damage.** Lipidic cubic phases (LCP) are an ideal delivery medium for serial crystallography because of the simple sample preparation, the compatibility with both soluble[20] and membrane proteins[3] and an easily controllable extrusion by high-viscosity injectors[3]. Similar to the search for suitable cryo-protectants in classical cryo-crystallography, different crystals may tolerate the embedding process differently. To solve this problem, a variety of different viscous media have been described[17, 21, 22]. In our hands, all crystals tolerated LCP very well and hence we only used this well-established medium in the present study. In our SMX setup (Fig. 1) we use nozzles between 50 and 75 μm, which is sufficient to avoid clogging by crystals, as larger crystals were usually crushed during mixing with a standard syringe coupler[23], without indications for loss of diffraction quality in our cases.

In optimization experiments with lysozyme crystals of an average size of  $\sim 10 \times 10 \times 10$–20 μm$^3$, we determined optimal beam parameters for data collection (see Supplementary Table 1). To test if a smaller beam is beneficial for data quality, we collected three data sets with three different foci of $20 \times 5$ μm$^2$, $10 \times 5$ μm$^2$ and $5 \times 5$ μm$^2$ with similar overall flux and flux densities of $0.4 \times 10^{10}$ ph/ (s × μm$^2$), $0.8 \times 10^{10}$ ph/ (s × μm$^2$) and $1.5 \times 10^{10}$ ph/ (s × μm$^2$), respectively. Using a data set collected in 20 min with the $5 \times 5$ μm$^2$ beam allowed us to determine the structure of lysozyme with data extending to 1.5 Å resolution (see Table 1 and Supplementary Fig. 1). The smaller $5 \times 5$ μm$^2$ beam resulted in the best data characteristics, with slightly higher overall signal-to-noise ratio and better CC*[24] in the high-resolution shell (see Supplementary Table 1). However, the effect of the smaller beam was not very pronounced, resulting only in an 18% increase in overall I/σ and a 0.09 Å increase in resolution (based on a CC* >0.5 criterion) as compared with the largest beam tested, despite the flux density being reduced to a quarter in this experiment (see Supplementary Table 1). Based on these results we decided to use the $20 \times 5$ μm$^2$ beam for most of the study as crystals from our other three proteins were of comparable size and it allowed us to use the full intensity beam. In some runs, we varied the longer side of the beam to adjust hit rate and signal to noise depending on the specific characteristics of the sample. The wider beam perpendicular to the flow of the LCP extrusion allowed scanning a larger volume of the crystal-carrying medium. Generally, this led to increased hit rates and thus balanced sample consumption, data collection speed and signal-to-noise ratio.

The speed of extrusion, the average crystal size, the beam shape and the applied flux determine the average radiation dose per crystal (see online methods and Supplementary Data 1). Hence, we adjusted the extrusion speed to 250 μm/s for most cases while using the maximum available photon flux (see Table 1) to balance strong diffraction vs. radiation dose and shorter time for data collection.

Data were collected continuously using the EIGER 16 M detector at 50 Hz. Instead of recording only one image per crystal the low jet speed, a thin beam and a frame rate of 50 Hz allowed each crystal being traversed by one beam width of 5 μm in one frame to avoid recording of multiple frames from the same crystal region.

Using eight runs from our A$_{2A}$ receptor 6 keV data with a hit rate of ~40%, we compared unbinned 50 Hz data with twofold binned 25 Hz data and fivefold binned 10 Hz data (see Supplementary Table 2). As in the original 50 Hz data, diffraction from the average sized crystal spans about six images (see Supplementary Fig. 2), the 10 Hz binned data nicely demonstrate the case where on average one diffraction pattern is collected per crystal. Comparison of data sets revealed that scanning a crystal with higher frame rate data collection for SMX is superior to lower frame rate data collection. This was supported not only by anomalous peak heights (see Fig. 2) but also by other data statistics such as signal-to-noise ratio in the higher resolution shells (see Supplementary Table 2). Using the anomalous signal to assess data quality, we show that the improved statistics are not owing to multiple patterns collected from a single crystal, which may bias statistics based on CC* (see Fig. 2). Thus, there are clear advantages of the latest detector generation with high frame rate and negligible dead time over slower detectors for SMX data acquisition.

Using the described data collection scheme, we observed 1–8 diffraction patterns per crystal (see Supplementary Fig. 2) corresponding to overall doses between 30 and 480 kGy per entire crystal, but maximally 60 kGy for each exposed crystal region (Table 1) as calculated with RADDOSE3D[25]. Such dose generally does not lead to a strong loss off diffraction[26]. Consequently, we did not observe site specific damage features such as broken disulfide bonds or decarboxylation of residues in all four case studies.

**Sample optimization.** To examine the time needed to adapt protein crystallization and crystal preparation for SMX, we selected the soluble MOSTO complex whose crystallization was previously optimized for classical crystallography[27]. Room-temperature structure determination of this complex is challenging because of the unusually high metal content, which makes it susceptible to radiation damage. After a simple optimization step, we were able to produce numerous microcrystals suitable for SMX in hanging drops (Supplementary Fig. 3). After embedding these crystals into LCP, we collected 68,788 indexable diffraction patterns in ~6 h with much less time necessary to collect a fully converged data set (see Fig. 4). The data extended to 1.7 Å resolution, and the resulting electron density map is of excellent quality (see Fig. 3a and b). Clearly, high-resolution structure determination at room-temperature by SMX can be fast and straightforward.

**Ligand soaking.** Next, we explored the suitability of SMX for the method that is inarguably responsible for most of the structures deposited in public or corporate databases: ligand soaking. We used microcrystals of the TD1 complex[28], and soaked them with the microtubule-destabilizing agent colchicine—a common medication against gout. The colchicine-binding site is the target for a new class of chemotherapeutic agents[29]. After a simple sample preparation (see materials and methods) and an incubation of 2 h, we collected 62,245 indexable diffraction patterns in ~9 h (see Table 1). The diffraction data extended to 2.1 Å (Table 1), and the ligand was clearly resolved and bound with full occupancy (see Fig. 3d). Colchicine showed the same binding mode as described previously[30, 31]. For comparison, we further collected a ligand free data set containing 66,271 images in 8 h. Useful data extended up to 2.1 Å resolution resulting in a high-quality electron density (see Fig. 3e). Despite using a much lower dose per crystal, the resolution of our TD1$_{col}$ SMX structure with 2.1 Å is only slightly lower than the 1.8 Å structure of TD1$_{apo}$ obtained by cryo-crystallography from a larger crystal (see Table 1 and Supplementary Table 4).

**Table 1 Data statistics**

| Data set | Lysozyme | A$_{2A}$R SMX | A$_{2A}$R Cryo | A$_{2A}$R 19 h 6 keV | A$_{2A}$R SFX | TD1$_{Apo}$ | TD1$_{Col}$ | MOSTO |
|---|---|---|---|---|---|---|---|---|
| X-ray energy (keV) | 12.4 | 12.4 | 12.4 | 6.0 | 9.5 | 12.4 | 12.4 | 12.4 |
| Measurement time (h) | 0.3 | 6.6 | ~ 4 | 19.4 | 0.36 | 7.7 | 8.6 | 5.9 |
| Nozzle size (µm) | 50 | 50 | – | 50 | 50 | 50 | 75 | 75/100 |
| Beam size (µm) | 5 × 5 | 20 × 5 | 20 × 5 | 15 × 5 | 1 × 1 | 20 × 5/10 × 5 | 10 × 5 | 20 × 5/40 × 5 |
| Flux (ph/s) | 3.9 × 10$^{11}$ | 1.5 × 10$^{12}$ | 1.5 × 10$^{11}$ | 4 × 10$^{11}$ | 2.4 × 10$^{11a}$ | 1.5 × 10$^{12}$/ 0.7 × 10$^{12}$ | 0.7 × 10$^{12}$ | 1.5 × 10$^{12}$ |
| Frame rate | 50 Hz | 50 Hz | 10 Hz | 50 Hz | 120 Hz$^b$ | 50 Hz | 50 Hz | 50 Hz |
| Crystal size (µm$^3$) | 15 × 10 × 10 | 30 × 30 × 5 | 30 × 30 × 5 | 30 × 30 × 5 | 30 × 30 × 5 | 15 × 10 × 10 | 15 × 10 × 10 | 50 × 20 × 20 |
| Dose per imagecrystal (kGy) | 10 | 30 | 6,000–13,000 | 13$^c$ | 6760 | 30/30 | 60 | 30/30 |
| Oscillation range | – | – | 0.1 | – | – | – | – | – |
| Jet speed (µm/s) | 250 | 250 | – | 250 | 1747 | 250 | 110 | 250/125 |
| Space group | $P4_12_12$ | $C222_1$ | $C222_1$ | $C222_1$ | $C222_1$ | $P2_1$ | $P2_1$ | $P6_322$ |
| Unit cell (a, b, c in Å, β/γ in °) | 78.6, 78.6, 38.9, 90 | 40.3, 180.1, 142.7, 90 | 39.4, 179.6, 139.9, 90 | 40.3, 180.1, 142.7, 90 | 39.9, 179.2, 141.2, 90 | 74.0, 91.4, 83.6, 96.8 | 72.9, 85.0, 84.3, 97.3 | 117.4, 117.4, 233.4, 120.00 |
| Collected images | 58,000 | 1,180,705 | 3500 | 3,496,230 | 155,241 | 1,388,078 | 1,544,487 | 1,054,366 |
| Crystals used | – | – | 6 | – | – | – | – | – |
| Indexed patterns | 27,000 | 128,086 | 3500 | 186,688 | 3563 | 6,6271 | 6,2245 | 68,788 |
| Patterns indexed (%) | 46.5 | 10.8 | 100 | 5.3 | 2.3 | 4.8 | 4.0 | 6.5 |
| Resolution | 24.84–1.50 | 25.2–2.14 | 50.0–1.95 | 34.0–2.67 | 20.2–1.70 | 36.1–2.13 | 35.0–2.05 | 24.2–1.70 |
| Number of reflections | 9,140,532 | 31,065,416 | 306,759 | 58,802,388 | 1,325,959 | 26,000,036 | 12,155,884 | 84,306,711 |
| Number of unique reflections | 20,181 | 30,837 | 32,392 | 30,230 | 56,793 | 65,679 | 62,424 | 110,141 |
| Multiplicity | 452.9 (4.6) | 1007.4 (8.1) | 9.5 (2.0) | 1945.2 (684.7) | 23.3 (3.0) | 395.9 (21.8) | 194.5 (6.2) | 765.4 (113.8) |
| Completeness | 94.7 (49.23) | 99.4 (93.83) | 87.6 (61.7) | 100 (100) | 93.7 (53.6) | 100 (100) | 91.9 (49.5) | 100 (100) |
| I/ sigma | 8.35 (0.72) | 13.17 (0.70) | 10.36 (1.13) | 24.76 (3.57) | 2.93 (0.44) | 6.56 (0.74) | 5.19 (0.52) | 5.29 (0.32) |
| CC$^*$ | 0.99 (0.53) | 0.99 (0.47) | 0.99 (0.79) | 0.99 (0.35) | 0.99 (0.45) | 0.99 (0.56) | 0.99 (0.82) | 0.99 (0.83) |
| CC1/2 | 0.996 (0.17) | 0.99 (0.12) | 0.99 (0.46) | 0.99 (0.07) | 0.97 (0.11) | 0.99 (0.19) | 0.99 (0.51) | 0.99 (0.53) |
| Rsplit/ Rmeas | 5.54 (220.70) | 4.66 (211.76) | 16.8 (91.8) | 2.50 (182.03) | 17.94 (315.36) | 6.38 (190.09) | 7.85 (236.65) | 10.85 (331.03) |
| R$_{cryst}$/R$_{free}$ | 15.8/19.9 | 20.7/23.1 | 18.1/21.1 | – | 21.7/23.1 | 17.9/22.6 | 19.3/23.9 | 18.0/20.1 |
| Ramachandran favored/ allowed/outliers | 129/2/0 | 378/10/0 | 425/3/0 | – | 384/5/0 | 998/24/1 | 966/30/4 | 497/14/0 |
| RMSZ (bonds) | 0.45 | 0.25 | 0.29 | – | 0.25 | 0.51 | 0.51 | 0.45 |
| RMSZ (angles) | 0.62 | 0.40 | 0.42 | – | 0.40 | 0.64 | 0.66 | 0.59 |
| Average B-factor | 37.0 | 78.0 | 52.0 | – | 77.0 | 74.0 | 86.0 | 47.0 |
| PDB code | 5NJM | 5NLX | 5NM2 | – | 5NM4 | 5NQT | 5NM5 | 5O5W |

$^a$Photons per pulse
$^b$Repetition rate
$^c$Diffraction weighted dose is 0, maximum dose given

**Data convergence.** In order to assess how much data are minimally necessary to obtain a data set that is converged, we generated simulated annealing (SA) ligand omit maps and plotted their cross correlation to the ligand density calculated from the model vs. the number of images included in a data set (See Fig. 4). These plots illustrate that convergence does depend mostly on multiplicity, but also on other factors as TD1 ($P2_1$), A$_{2A}$R SMX ($C222_1$) and MOSTO ($P6_322$) converge at ~10,000, 1,000 and 2,000 indexed patterns corresponding to multiplicities of 21.3, 15.6 and 28.3, respectively. Based on the average indexing rate, data collection times for these converged data sets were 82, 3 and 10 min for TD1, A$_{2A}$R SMX and MOSTO, respectively. However, ligands already reach 90% of their final cross correlation to the generated SA omit maps after 4,000, 900 and 900 indexed patterns (indicated by dotted lines in the plot) corresponding to 35, 3 and 5 min of data collection for TD1, A$_{2A}$R SMX and MOSTO, respectively. Whether a ligand is bound can thus be already determined with relatively few images and after a short time of data collection. Interestingly, the SMX and SFX data for A$_{2A}$R converge following a similar trajectory although we observe a slightly higher final correlation with the ligand for data collected at the XFEL likely because of the higher overall resolution of these data.

**Native SAD phasing using SMX.** To demonstrate the suitability of SMX for de novo phasing, we choose the human A$_{2A}$ receptor, a G protein-coupled receptor. Although members of this class of intra-membrane receptors are common targets for pharmacological intervention, they are known to be difficult to crystallize. Furthermore, crystals of the A$_{2A}$ receptor decay quickly in room-temperature experiments[32], making these crystals ideal to demonstrate the suitability of SMX for difficult targets.

We collected a large data set of the A$_{2A}$ receptor at a photon energy of 6 keV where the anomalous signal is prominent,

and X-ray absorption effects can still be neglected. Structure solution was attempted in parallel to data collection and already succeeded during 19 h of beamtime (Fig. 6). SHELXD identified 18 sites with CCall of 25.5 and CCweak of 11.9, which were clearly separated from non-solutions (Fig. 5b). SHELXE phasing, density modification and chain tracing resulted in a good separation of hands after 10 cycles (Fig. 5c). Automatic model building of the A$_{2A}$ receptor on the 6 keV data alone improved initial phases, resulting in a map of sufficient quality for manual improvement with a model map CC of 60% to the final model.

Criteria for the identification of a diffraction hit were chosen conservatively, and by using different hit finding criteria as well as peaks identified by CHEETAH for indexing we could retrospectively solve the structure with data collected in 5 h (Supplementary Table 3, Fig. 6d and methods).

SHELXD identified 14 sites with CC$_{all}$ of 25.5 and CC$_{weak}$ of 9.3 and clearly separated solutions from non-solutions (Fig. 5e). It was necessary to perform 100 cycles of SHELXE phasing, density modification and chain tracing to obtain a good separation of hands (Fig. 5f). Automatic model building of the A$_{2A}$ receptor on the 2.7 Å data collected at 6 keV alone improved initial phases, resulting in a map of sufficient quality for manual improvement of the model with a model map CC of 55% to the final model. Overall, native-SAD phasing was possible using only ~10 µl LCP containing 12 mg ml$^{-1}$ A$_{2A}$ receptor. The high efficiency of high-viscosity injectors together with our high frame rate serial-scanning approach, thus allows to target proteins for which only such submilligram quantities are readily available, even in the case of a challenging membrane protein.

**A$_{2A}$ receptor structures determined by three methods.** In addition to the long wavelength data collected for native SAD phasing we collected a data set at 12.4 keV to obtain higher resolution. We collected 128,086 indexed diffraction patterns in

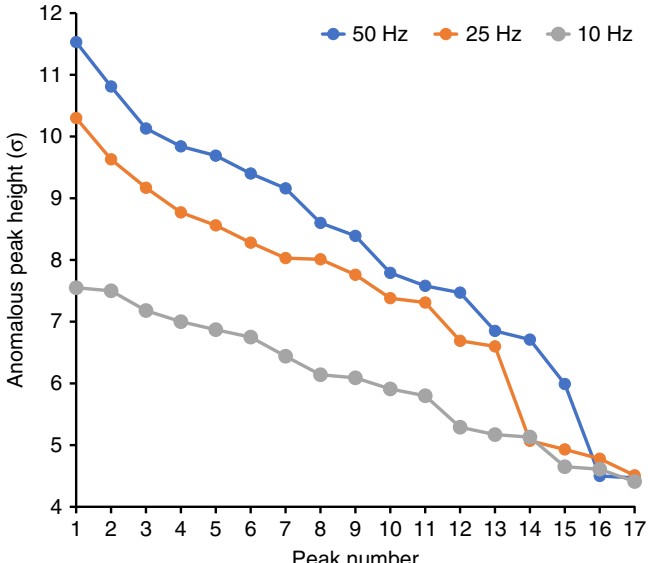

**Fig. 2** Detector frame rate vs. anomalous peak height. The plot reveals that high frame rate data collection results not only in larger anomalous peak heights but also resolves more sites

6.6 h with data extending to 2.1 Å resolution. The fully converged data set shows a well-defined electron density (See Fig. 6). For comparison, we also collected a more conventional cryo-crystallography data set of the remaining crystals at the same beamline. The final cryo-data set was merged from six crystals selected from a large number identified by extensive grid scanning. All in all, 350° of fine sliced data (corresponding to 3,500 diffraction patterns) were collected in ~4 h and the data extended to 2.0 Å resolution. In order to compare these synchrotron data to data collected at FEL sources, we also measured a small data set at Linac Coherent Light Source (LCLS). In 22 min that were available for data collection, we obtained 3563 indexable diffraction patterns resulting into data extending up to 1.7 Å resolution. Despite the small number of images collected, the data had already converged (see Fig. 4). Upon visual comparison of electron density maps, the resolution difference between all data sets is subtle, showing almost no difference between cryo- and SMX data apart from weaker density around water molecules in case of the SMX data (See Fig. 6). The electron density map obtained from LCLS data is of the best quality and shows slightly better SA omit map cross correlation for the ZM241385 ligand (see Fig. 4).

## Discussion

Time-resolved crystallography as well as modern approaches to study the conformational variability of proteins rely on room-temperature structure determination, which allows us to gain detailed understanding of protein functionality. Room-temperature structure determination with the rotation method has one fundamental problem: radiation damage will always limit the maximum attainable resolution, and this is especially true for small crystals from pharmaceutically interesting membrane proteins such as G protein-coupled receptors. Injector-based serial crystallography is a convenient way to distribute much larger doses over thousands of crystals. We demonstrate this approach using microcrystals of the human $A_{2A}$ receptor for which we collected data to 2.14 Å resolution with our SMX method and to 1.95 Å for the same crystals using cryo-crystallography. Of course, one major advantage of FEL sources is that one can avoid such a maximal dose limit by the diffraction-before-destruction principle[33]. Hence, we were able to

measure data to up to 1.7 Å resolution for the $A_{2A}R$ even with only a very short data collection at LCLS.

Our current SMX method combines the strengths of our previous SMX approach that was demonstrated on Bacteriorhodopsin[34], namely short exposure times to mitigate crystal motions and a small beam to improve the signal-to-noise ratio, with the benefits of continuous shutterless data collection and a broader beam to overcome the problem of slow data acquisition. As opposed to another shutterless approach demonstrated on the PILATUS detector with lysozyme as a model system[17], we collected multiple high frame rate-patterns instead of a single diffraction pattern per crystal, used a much smaller beam and the opposite beam geometry ($20 \times 5 \, \mu m^2$ instead of $10 \times 30 \, \mu m^2$). Using 50 Hz frame rates as made possible by high frame rate detectors such as EIGER and PILATUS3, we were able to counteract the negative impact of crystal movement during exposure of each individual frame as realized by using the shutter in our previous work. High frame rate data collection also avoids empty exposure of indexable frames after crystals already passed the beam. This leads to an increased signal-to-noise ratio by reducing the recorded background (see Supplementary Fig. 4).

In our experiments, the jet speed and detector readout were ideally configured such that although one detector frame was recorded, the extruded medium (and with it the crystals) moved one beam width. This allowed for collection of multiple diffraction patterns per crystal, without repeatedly exposing the same portion of the crystal. In case of the $A_{2A}R$ we used a 5 μm-thin beam and crystals traversed 5 μm during collection of one pattern (see Table 1 and Supplementary Data 1). This setup improved data collection statistics significantly. The improvement of data statistics (see Supplementary Table 2) by collecting multiple diffraction patterns per crystal is supported by dramatically increased anomalous signals (see Fig. 2), excluding bias as a source of statistics improvement.

At the same time, keeping the broadest dimension of the beam perpendicular to the direction of crystal travel minimized sample consumption and improved hit rates in our experiments. In the case of MOSTO, where we initially suffered from low hit rates, we used a very wide but thin $40 \times 5 \, \mu m^2$ beam. In combination with a larger nozzle size, a larger volume could be "scanned", allowing to collect a large data set within 6 h of beamtime while still obtaining data to high resolution.

We call this approach "crystal scanning", because it balances the hit rate and the signal-to-noise ratio by scanning across the extruded viscous carrier medium with a wide but thin beam (see Supplementary Fig. 4). In addition, it maximizes the amount of unique diffraction patterns by scanning crystals with a high frame rate that is matched to the crystal movement speed and the beam dimension in the direction of crystal movement. The fast readout of the EIGER detector and its duty cycle better than 99% [19] are crucial for this strategy. At the frame rates we used, also PILATUS3 detectors with a duty cycle of ~95% at 50 Hz would appear to be suitable for the strategy presented here, although the higher deadtime, which is not evenly spread-out over the individual frames, may lead to a slight decay in data quality. Our data collection scheme is very well suited for future time-resolved experiments, because it can combine the ultra-high frame rates of the EIGER detectors (up to 3,000 Hz) with laser excitation to achieve time resolutions in the millisecond range and beyond. Other approaches like choppers[35], single bunch operation[36] or precisely timed and short-pulsed lasers are able to increase achievable time resolutions even further, but the time resolution obtained simply by detector readout speeds will allow to collect whole-time series in a single experiment. To allow processing of high frame-rate data, several images can be merged together, resulting in lower frame rates but stronger diffraction patterns to

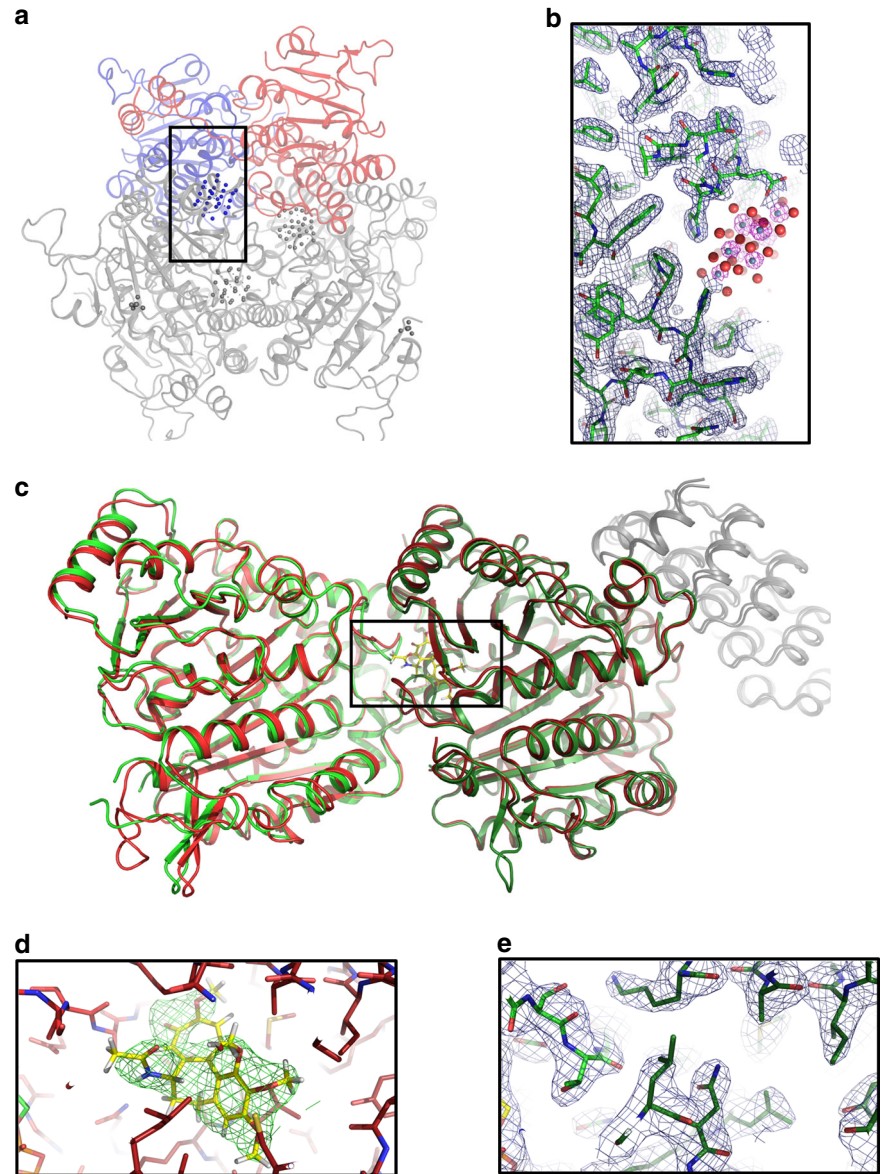

**Fig. 3** Structures of the molybdenum storage protein (MOSTO) and Tubulin (TD1). **a** The protein forms a heterohexameric $(\alpha\beta)_3$ cage-like structure with polyoxomolybdate clusters bound inside the cage. The α- and β- subunits present in the asymmetric unit of the SMX structure are shown as red and blue cartoons and the remaining two αβ-units completing the heterohexamer are shown in *gray*. **b** The zoomed polyoxomolybdate cluster binding site was drawn for the polypeptide with an $2F_o$–$F_c$ electron density in *blue* at 1σ and for the molybdenum cluster (molybdenum and oxygen shown as *gray* and *red spheres*) in *pink* at 10σ, highlighting the positions of the polynuclear molybdenum-oxide aggregates. **c** Overlay of $TD1_{apo}$ and $TD1_{col}$ SMX structures shown as *green* and *red* cartoons, respectively. The DARPin is shown as *gray* cartoon and α-tubulin and β-tubulin subunits are shown as *bright red* or *bright green* and *dark red* or *dark green*, respectively. **d** Zoom in on $TD1_{col}$. Surrounding residues are shown as *red sticks* and colchicine is shown as *yellow sticks* with the surrounding $F_o$–$F_c$ density of the structure refined without the ligand at 3σ. **e** Zoom in on $TD1_{apo}$. Surrounding residues are shown as *green sticks* with the surrounding $2F_o$–$F_c$ density of the structure at 1σ

allow indexing, integration and background subtraction. Leaving out individual images from the merged images and using the Hadamard transform[37] to deconvolute them, may allow time resolved analysis far beyond indexable individual diffraction patterns at these frame rates. With the advent of single shot serial crystallography, data processing progressed significantly by improving detector geometries[38], as well as including prediction- and post-refinement[39–41]. We expect these developments to continue at their fast pace, allowing challenging time resolved experiments to be even more feasible.

Using this strategy, we were able to solve the structure of a human membrane protein pharmaceutical target, the $A_{2A}$ receptor, by native-SAD phasing, which demonstrates that even

high quality data necessary for native-SAD phasing are obtainable in 5 h of beamtime with injector-based SMX. This was possible on very small crystals $(30 \times 30 \times 5\,\mu m^3)$ and at room-temperature. Based on reports of gadolinium based[42, 43] and native sulfur/chloride SAD phasing at XFEL sources[44], phasing exploiting other anomalous scatterers than sulfur should be even easier to implement. At the synchrotron, both the data collection time and the number of indexed images were lower than those required for phasing of the $A_{2A}$ receptor at the LCLS[6]. Reasons for this unexpected finding may include (i) the use of larger crystals in our experiment at the Swiss Light Source (SLS), (ii) lower partiality of individual reflections due to slight rotation of crystals during exposure at the synchrotron, (iii) shot to shot

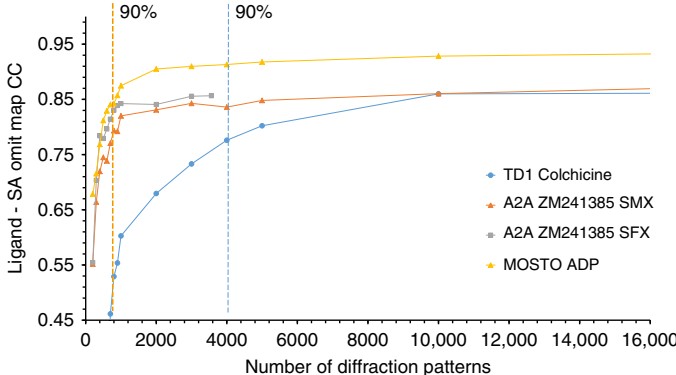

**Fig. 4** Data convergence. The plot shows the cross correlation between the final refined positions of soaked colchicine bound to TD1, co-crystallized ZM241385 bound to $A_{2A}R$ for SMX and SFX data and intrinsic ADP bound to MOSTO with the $2F_o–F_c$ simulated annealing ligand omit map vs. the number of indexed diffraction patterns. The dotted lines mark the spot were 90% of the final cross correlation is reached

variation in the spectrum and intensity of XFEL pulses, as well as (iv) the higher dynamic range of the single photon counting EIGER 16 M detector[19] vs. the integrating CSPAD[45].

Interestingly, we observe unexpected differences between cryo- and room-temperature structures of tubulin. Visualizing protein backbone mobility by ensemble models[46] (see Methods), we observed overall similar patterns of mobility in different parts of the structures, with overall increased mobility of loop regions for room-temperature structures, as expected. However, for the colchicine unbound structure, the T7 loop of β-tubulin that shields ligands bound to the colchicine site from the solvent and the α-tubulin T5 loop, which is also involved in ligand binding to the colchicine site, show an opposing mobility pattern in both structures. The T7 loop appears flexible and the T5 loop more rigid in the room-temperature structure, whereas the cryo-structure seems to exhibit a more mobile T5 loop and a more rigid T7 loop (Supplementary Fig. 5 and Supplementary Table 4). This observation indicates that apart from allowing time resolved experiments, room-temperature structures may provide additional information that could help to understand ligand binding and protein dynamics.

When comparing cryo-crystallography, SMX and SFX, we observed an immediate resolution advantage of XFEL over synchrotron crystallography (Table 1 and Fig. 6). Clearly, one advantage of XFELs is the ability to deliver and diffract photon amounts that will destroy any crystal in a synchrotron experiment. This advantage is most relevant when small, weakly diffracting crystals are studied or in cases where short pulses are needed to collect time-resolved data. Nevertheless, XFEL beamtime will not be available for routine experiments in the foreseeable future and synchrotron based SMX taps into the large pool of available synchrotron beamlines. The upcoming diffraction-limited storage rings and specialized beamlines will improve on flux densities and instrumentation. They will further reduce the time needed for successful SMX experiments and close the already small resolution gap between cryo-crystallography and practically damage free room-temperature structure determination by SMX.

Information on whether a ligand is bound before collecting a high-quality data set is important for structure-based drug design approaches like fragment screening, where many structures of a target protein with a variety of small molecules have to be determined. In our experiments data collection times of 3–35 min were enough to reach 90% of the final cross correlation between SA omit maps and the refined ligands (see Fig. 4). Liquid handling is easier to automate than crystal fishing, freezing and mounting of crystals in LCP and initial ligand binding can be

detected within minutes when using SMX. Hence, SMX is particularly well-suited for automated crystallographic fragment screening and other structure based screening approaches.

In conclusion, we show that SMX at synchrotrons allows fast, straightforward structure determination at room-temperature for large soluble macromolecular complexes as well as membrane proteins. Using SMX, a much larger dose can be distributed over many crystals, resulting in higher resolution structures with less-radiation damage compared with classical room-temperature methods. Collecting data using modern fast frame rate detectors produces results of excellent quality, making even native SAD phasing possible in less than a single 8-h synchrotron shift. Using ultra high frame rate detectors and next-generation diffraction-limited sources, time-resolved measurements in the micro- and perhaps nanosecond range may become possible at synchrotrons. Given the comparatively simple sample preparation, data collection, data processing and its great potential for automation, we believe that SMX is the method of choice for room-temperature structure determination and fragment screening approaches.

## Methods

**Lysozyme sample preparation.** Chicken egg white Lysozyme from Alfa Aeser (J60701) was dissolved at a concentration of 50 mg ml⁻¹ in 100 mM Na Acetate pH 3.0. Of this lysozyme solution, 500 µl were mixed with 500 µl of precipitant, 19.04% NaCl, 5.44% PEG 8,000, 68 mM Na Acetate pH 3.0 in an Eppendorf tube and incubated overnight at 20 °C. Crystals grown in batch were spun down at room-temperature with 2,000 g for 1 min. Almost all supernatant was removed and crystals were re-suspended using a 200 µl pipette. Crystals were then pipetted into the back of a Hamilton syringe and mixed with ~70% monoolein through a 400 µm LCP coupler at room temperature until the cubic phase was formed.

**TD1 sample preparation.** Synthetic DNA encoding for the DARPin D1[28] was cloned into a pET-based expression vector (see Supplementary Table 5) and expressed in *Escherichia coli* BL21 DE3 (purchased from Stratagen). The DARPin protein was purified by Ni-affinity chromatography and size exclusion chromatography. Bovine brain tubulin was purchased from the Centro de Investigaciones Biológicas (Microtubule Stabilizing Agents Group), CSIC, Madrid, Spain. The TD1 was formed by mixing tubulin and Darpin 1 in a 1:1.1 ratio and concentrated to 20 mg ml⁻¹ before crystallization. TD1 was crystallized by hanging drop vapor diffusion at 20 °C in 20–24% PEG 3350, 0.2 M Ammonium Sulfate, 0.1 M Bis-Tris Methane pH 5.5, mixing 2 µl of the complex with 2 µl of precipitant solution. On the next day, a few TD1 crystals were observed in each drop. To reach a larger number of crystals 1 µl water was then added and the drop mixed a few times. After another night incubation at 20 °C, each crystallization drop contained numerous TD1 microcrystals.

SMX sample preparation. Individual hanging drops were pooled by pipetting into an Eppendorf tube, the glass covers were carefully washed with mother liquor. Crystals were then spun down at room-temperature with 2,000 g for 1 min. Almost all supernatant was removed and crystals were re-suspended. Crystals were then pipetted into the back of a Hamilton syringe and mixed with ~ 70% monoolein through a 400 µm coupler at room temperature until the cubic phase formed.

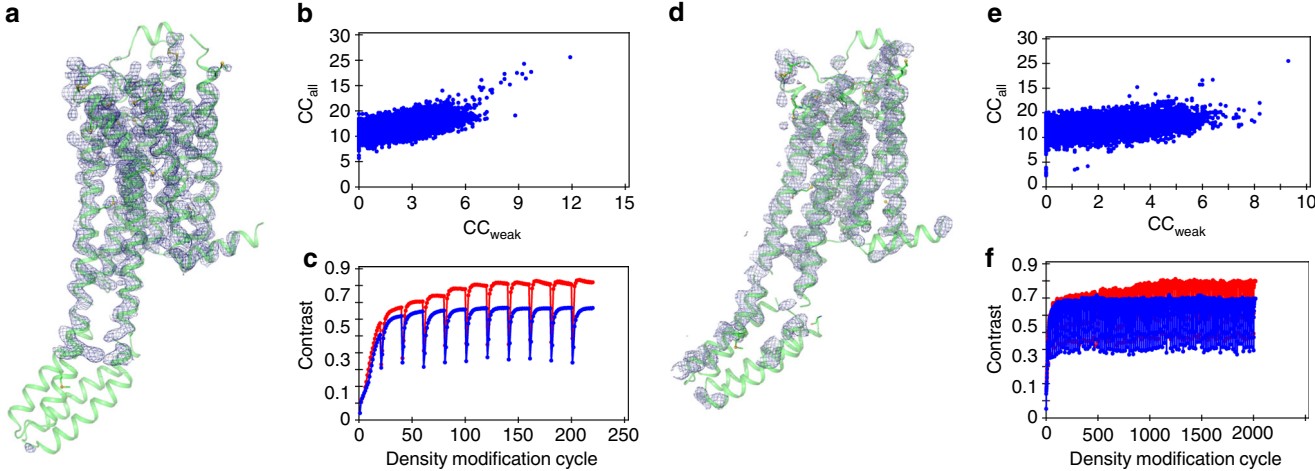

**Fig. 5** Native SAD phasing of the human $A_{2A}R$ with high quality (19 h data collection) and minimal (5 h data collection) data sets. **a** Experimental density of the high-quality data set after SHELXE phasing, density modification and chain tracing superimposed with the refined structure. **b** Substructure solution of the high-quality data set in SHELXD. **c** Combined phasing, chain tracing and density modification of the high-quality data set in SHELXE. **d** Experimental density of the minimal data set after SHELXE phasing, density modification and chain tracing superimposed with the refined structure. **e** Substructure solution of the minimal data set in SHELXD. **f** Combined phasing, chain tracing and density modification of the minimal data set in SHELXE

**Ligand soaking**. A mixture of 6 µl 100 mM colchicine in 100% DMSO, 6 µl of TD1 crystallization buffer (22% PEG 3350, 0.2 M ammonium sulfate, 0.1 M bis-tris methane pH 5.5) and 18 µl Monoolein was prepared to a final concentration of 20 mM colchicine and 20% DMSO. The protein crystals were prepared as described above. A total of 5 µl ligand mixture was then mixed 100 times with 22 µl crystals dispersed in LCP using two Hamilton syringes and a standard 400 µm coupler, resulting in final concentrations of 3.7 mM colchicine and 3.7% DMSO. Crystals were incubated for 2 h before data collection to allow colchicine binding.

**MOSTO sample preparation**. MOSTO was heterologously produced in *E. coli* BL21 (DE3) cells with a combination of vectors harboring different molecular chaperones[47] (available on request from Professor Bernd Bukau (ZMBH, University of Heidelberg, Neuenheimer Feld 282, D-69120 Heidelberg, Germany))[48]. The protein was purified via strep-tag affinity chromatography followed by ion exchange and size exclusion chromatography (50 mM MOPS; 50 mM NaCl pH 6.5). Prior to crystallization the protein was concentrated to 25 mg ml$^{-1}$ and 1 mM ATP, 1 mM MgSO$_4$ and 1 mM Na$_2$MoO$_4$ was added. Standard hanging drop crystallization conditions (1 M NH$_4$H$_2$PO$_4$, 0.1 M sodium citrate pH 5.6 and 20% Glycerol) at 18 °C were modified to decrease the size of the crystals (compare Supplementary Fig. 5). First, a 24-well screening plate was set up with increasing concentrations of phosphate ranging from 1.1 M to 1.7 M NH$_4$H$_2$PO$_4$. From this plate, optimal conditions yielding sub-50 µm crystals were found to be 0.1 M sodium citrate pH 5.6, 1.4 M NH$_4$H$_2$PO$_4$ and 20% glycerol as reservoir solution. Subsequent 24-well plates for crystal production were prepared with a 2 + 2 µl drop size. For SMX individual hanging drops were pooled by pipetting into an Eppendorf tube, the glass covers were carefully washed with mother liquor. Crystals were then pipetted into the back of a Hamilton syringe and mixed with ~ 70% monoolein through a 400 µm LCP coupler at room temperature until the cubic phase was formed.

**A$_{2A}$ receptor sample preparation**. The A$_{2A}$ receptor construct used in this study contains eight thermostabilizing mutations, A54L, T88A, R107A, K122A, L202A, L235A, V239A and S277A, and another mutation N154A to remove a glycosylation site. A FLAG tag has been added to the N-terminus and after K315, residues corresponding to the A$_{2A}$ receptor have been replaced by three alanines and 10 histidines. Apocytochrome b 562 RIL (bRIL) is inserted into the third intracellular loop (ICL3) between residues L208 and E219.

The construct was expressed in the baculovirus system and purified[49] as follows: membranes containing overexpressed A2A receptor were resuspended in 40 mM Tris-HCl pH 7.6, 1 mM ethylenediaminetetraacetic acid supplemented with ethylenediaminetetraacetic acid-free protease inhibitor cocktail (Roche) and stored at −80 °C until use. To purify A$_{2A}$R, membranes were thawed and solubilized in 1.5% (w/v) n-decyl-β-D-maltopyranoside (DM) for 1 h at 4 °C in the same buffer supplemented with 5 µM ZM241385 and insoluble debris spun down by ultracentrifugation. A$_{2A}$R was purified using Ni-NTA superflow resin (Qiagen) in 40 mM Tris pH 7.4, 200 mM NaCl, 0.15% DM and 5 µM ZM241385, washed with 25CV then eluted with 280 mM imidazole in the same buffer. Fractions containing A$_{2A}$R were pooled, concentrated with Amicon Ultra-15 centrifugal device with 50 K MWCO and loaded onto a 10/30 S200 size exclusion column (GE Healthcare) in 40 mM Tris pH 7.4, 200 mM NaCl, 0.15% DM, and 5 µM ZM241385 at 0.5 ml/min. Fractions containing A$_{2A}$R were pooled and concentrated to ~ 30 mg/ml for

crystallization trials. Receptor purity and monodispersity were analyzed by SDS-PAGE and analytical gel filtration. Protein concentration was determined at 280 nm using the protein extinction coefficient of 58745 M$^{-1}$/cm on a Nanodrop 8000 UV–vis spectrophotometer (Thermo Scientific). Purified A$_{2A}$ receptor was concentrated to 25–30 mg/ml, mixed with monoolein containing 10% (w/w) cholesterol at a protein to lipid ratio of 1:1.5 (w/w). Crystals of the A$_{2A}$ receptor were directly grown in LCP containing gas-tight syringes with 5 µl LCP injected into 50 µl precipitant solution (0.1 M sodium citrate pH 5.0, 0.05 M sodium thiocyanate, 28–34% PEG400, 5 mM ZM241385, 2% (v/v) 1,6-hexanediol). After the crystals formed within the LCP volume, a syringe coupler was attached and the precipitant solution was slowly removed by gently pushing on the syringe plunger. This step was necessary to retain all LCP with crystals with a minimal amount of precipitant. Step-wise addition of 3 µl molten monoolein followed by through mixing was repeated at room temperature until a clear and homogeneous LCP was restored.

**Experimental setup**. A schematic drawing of the installation at the X06SA beamline at the Swiss Light Source is shown in Fig. 1. The high-viscosity injector (purchased from Professor Uwe Weierstall, Arizona State University) was installed at Beamline X06SA at SLS. We pre-installed the required HPLC tubing and the helium gas lines to the beamline to allow the setup to be changed from cryo-crystallography to SMX within 1 h and facilitate routine use of the method. The LCP medium containing the crystals was extruded perpendicular to the ground level through 50, 75 and 100 µm nozzles. The speed of extrusion ranged from 110 to 500 µm/s and the jet was stabilized using He gas passed through the gas aperture. The applied gas pressures typically ranged between two and four bars. Pressures up to 40 bars were used to clean the nozzle from sample that was not extruded properly. Extruded sample was caught in a sample catcher connected to a membrane pump.

**Data collection**. Data were collected with a continuous beam using an EIGER 16 M detector at frame rates indicated in Supplementary Table 1. The widest dimension of the beam was horizontal to the ground level (see Supplementary Fig. 1). Data were collected in separate runs containing maximally 100,000 images each.

**Data processing**. Most data were processed using CrystFEL version 0.6.2[50] directly on the acquired images. Images were pre-selected by our live monitoring software that implemented Cheetah[51] peak finding algorithm 8 for peak detection. Parameters used during the beamtime were: signal to noise ratio = 10, minimum number of pixels per peak = 3 and minimum number of spots 10. For processing optimization, data with minimum SNR = 5 and minimum number of peaks = 20 were used. The final processing with optimized parameters was performed on all images. In this approach, all peaks used for indexing were identified using the Zaeferer algorithm as implemented in CrystFel. We choose this approach for simplicity and to avoid unnecessary data duplication.

For 10 runs of the A$_{2A}$ receptor 6 keV data (corresponding to 5 h of data collection time) we used the EIGER specific implementation of Cheetah[51] developed by Takanori Nakane (https://github.com/biochem-fan/cheetah/tree/eiger) and subsequently used the Cheetah found peaks for indexing.

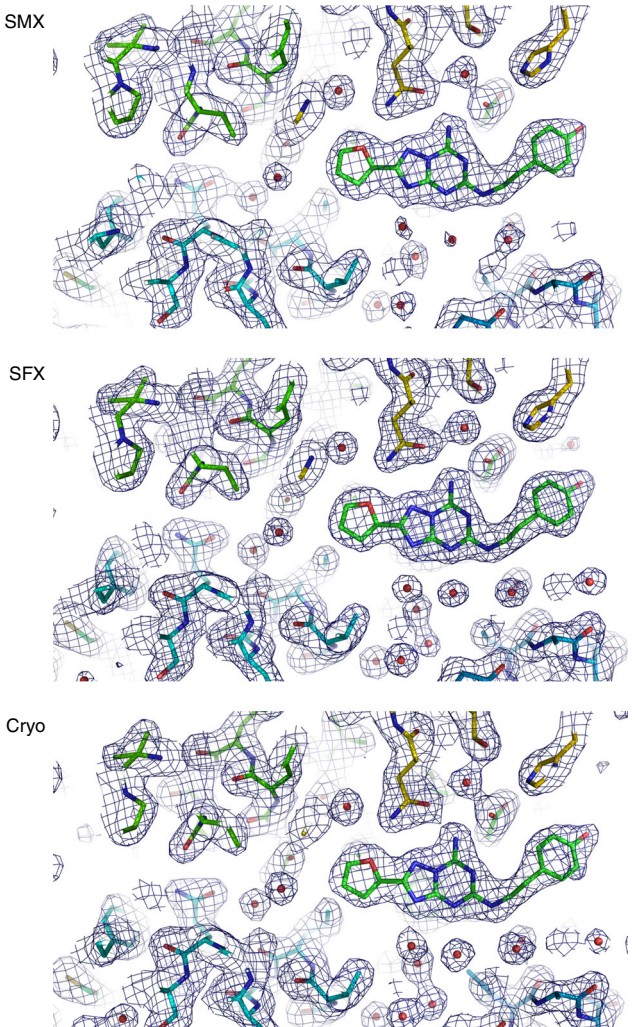

SMX

SFX

Cryo

**Fig. 6** Comparison of the $A_{2A}$ receptor electron density obtained using SMX, SFX and conventional cryo-crystallographic data. The structures show the ligand binding pocket with $2F_o-F_c$ maps shown at $1\sigma$. All data sets were collected using $A_{2A}R$ crystals prepared in the same way to facilitate a direct comparison of the techniques. The SMX density is well defined, but the overall resolution (2.1 Å) is lower than in SFX (1.7 Å) and cryo-crystallography (1.95 Å)

After indexing and integration using indexamajig, data were scaled without a partiality model in partialator, applying a per pattern resolution cutoff in the process. Data were then converted into mtz files and intensities were converted to structure factor amplitudes by using truncate. All data were kept throughout the process but for the reported statistics and refinement were cut at either CC* falling below 50% or the completeness dropping below 50%.

**Cryo data collection and processing for the $A_{2A}$ receptor.** To collect classical crystallography data, we used crystals originally prepared for SMX to allow comparison between the two techniques. Crystals were harvested and cryo-cooled in liquid nitrogen. The crystals were pre-screened and from the best crystals data were collected to obtain a complete data set. We needed overall 4 h of synchrotron beamtime to obtain a full data set merged from six of the best crystals, not counting the time for harvesting and cryo-cooling of the sample. Overall 350° of data from six different crystals were collected at beamline X06SA with a $20 \times 5\,\mu m^2$ beam and 10% transmission (see Supplementary Table 1). The average crystal used in the data set diffracted to ~2.2 Å with only one crystal diffracting to higher resolution. Data were processed using XDS[52].

**SFX data collection and data processing for the $A_{2A}$ receptor.** The SFX experiment on the $A_{2A}$ receptor was performed at the CXI experimental station[53]. The LCP stream from the $50\,\mu m$ injector nozzle was aligned with the path of 50 fs long X-ray pulses delivered at a repetition rate of 120 Hz. X-ray transmission was set to 10% to avoid Bragg peak saturation on the CSPAD detector, resulting in $10^{11}$

photons per pulse. The collected data were preprocessed with Cheetah[51] in order to select frames containing diffraction patterns. Selected images were indexed and integrated and merged using CrystFEL version 0.6.2[50].

**Calculation of radiation doses.** The radiation dose (diffraction weighted dose) was calculated using RADDOSE3D[25] with the parameters given in Supplementary Table 1, assuming no rotation of the crystals and choosing the smallest crystal dimension collinear to the direction of the beam, hence the values represent maximum doses. Crystal movement was taken into account by matching the exposure time to the amount of time the crystals traverse one full height of the beam (0.02 s in case of 250 µm/s jet speed and 5 µm beam height, see Supplementary Data 1). The dose applied owing to the Gaussian beam shape to the region above the beam, that was fully exposed in each consecutive frame was neglected in the dose calculations, since it is small compared with the dose received in the full exposure.

**Experimental phasing of the $A_{2A}$ receptor.** Initial phasing was achieved in parallel to data acquisition. Data processed with CrystFel[50] was converted into XSCALE format and passed through XSCALE[52]. Data preparation, sub structure identification and experimental phasing were performed using the SHELXCDE[54] pipeline with the graphical user interface hkl2map[55] to data preparation using SHELXC. The substructure was identified using SHELXD, using data up to 2.65 Å resolution and $E$-values to 1.3 in a search for 12 sites with expanding two sites into disulfides. Phasing was possible entirely on anomalous data, including data up to 2.6 Å resolution by performing 10 cycles of SHELXE with auto tracing and search for alpha helices enabled.

Phasing with the minimal amount of collected data was achieved by processing the images as described above. Data preparation, sub structure identification and experimental phasing were performed using the SHELXCDE pipeline with the graphical user interface hkl2map to data preparation using SHELXC. The substructure was identified using SHELXD, using data up to 2.75 Å resolution and $E$-values to 1.2 in a search for 11 sites without expanding into disulfides. Phasing was possible entirely on anomalous data collected up to 2.67 Å resolution by performing 100 cycles of SHELXE with auto tracing and search for alpha helices enabled (see Supplementary Fig. 5).

**Model building of the $A_{2A}$ receptor.** For the minimal data set (5 h data collection), Automatic model building of the $A_{2A}$ receptor was performed using phases of SHELXE and the 6 keV $A_{2A}R$ data directly resulting in 305 (68%) residues built and 142 residues sequenced after BUCCANEER[56], resulting in a map of sufficient quality for manual improvement of the model with a model map CC of 55% to the final model. For the high-quality data set (19 h data collection), automatic model building of the $A_{2A}$ receptor in Buccaneer was performed using phases of SHELXE and the 6 keV $A_{2A}R$ data directly resulting in 328 (73%) residues built and 209 residues sequenced resulting in a map of sufficient quality for manual improvement of the model with a model map CC of 60% to the final model.

**Refinement and model building.** Lysozyme: the structure was solved by molecular replacement with Molrep[57] using PDB entry 3LZT as model. The structure was then refined in refmac5[58] and finally in PHENIX[59], sidechains and water molecules were adjusted using Coot[60]. MOSTO. The structure was refined iteratively by refmac5 directly using PDB entry 4F6T and manual adjustment in Coot. Final refinement cycles were carried out using autoBUSTER (Global Phasing Ltd.). TD1 (Apo and Col): The structure was solved using PDB entry 4DRX and refined iteratively using refmac5 and iterative model building in Coot. The last cycles of refinement were carried out using autoBUSTER (Global Phasing Ltd.), in case of $TD_{Col}$ the colchicine ligand was refined using quantum mechanical description of the ligand. $A_{2A}R$ (SMX): We extended the phases obtained from the high-quality 6 keV data set on the 12.4 keV room temperature SMX data and used automatic model building in PHENIX followed by iterative refinement in refmac5 and manual model building in Coot. In the final stages, refinement was carried out using PHENIX. $A_{2A}R$ (SFX): The refined model from $A_{2A}R$ (SMX) was directly refined against the SFX data using PHENIX. $A_{2A}R$ (cryo): The structure was solved using PDB entry 5IU4 and subsequently refined using PHENIX.

**Simulated annealing omit maps/model map cross correlations.** The final processed streams of serial data were scaled and merged using partialator[50], using the '—stop-after' option to generate smaller data sets. For the resulting intensities, mtz files were generated, which were then converted into structure factor amplitudes using truncate[61]. The final refined model was then used to calculate SA (cartesian) omit maps omitting the ligands in PHENIX. The cross correlation was taken from the 'Map Correlation' task of CCP4[61].

**Ensemble refinements of TD1 structures.** For the room temperature ensemble refinements with and without the colchicine ligand we used $TD1_{col}$ and $TD1_{apo}$ data (see Table 1). For the ensemble refinement of the cryo-structure without colchicine we used data collected on a single TD1 crystal harvested before addition of water to produce micro-crystals that was cryo-protected using 25% PEG 400, 8%

PEG 3350, 0.2 M Ammonium Sulfate and 0.1 M Bis-Tris Methane pH 5.5. Data were collected at Beamline X06SA using a $20 \times 5\,\mu m^2$ beam using the Eiger 16 M detector, the structure of the refined room temperature structure TD1$_{apo}$ was directly refined in PHENIX against the data, with manual adjustments in Coot (see Supplementary Table 4). For the ensemble refinement of the cryo-structure with colchicine we used PDB entry 5EYP, replaced the α- and β- chains with the ones from our final refined room temperature model of TD1$_{col}$ but kept the DARPin, which had a slightly different sequence. In all cases crystal packing was almost identical, despite slight variations of unit cell constants.

All structures for the ensemble refinement were prepared in the same way. First, a model with no gaps regardless of observed density was generated, sidechains for which no density was observed were removed. Then a refinement over 10 cycles was carried out in PHENIX, optimizing stereochemistry and atomic displacement parameter (B-Factor) weights as well as updating waters. Sidechains were adjusted to fit the density in Coot. Afterwards TLS groups were determined automatically. Then, another refinement round with the automatically determined TLS groups was carried out, with the option 'Automatically add hydrogens to model' enabled, but solvent update disabled. The models generated in this way were then refined with a fraction of 0.8 of all atoms included in TLS fitting with ordered solvent update using PHENIX Ensemble refinement against data cut to 2.35 Å resolution. The resolution of the data was reduced to ensure the same amount of data for the refinement as well as data with CC$_{1/2}$ above 50% for all refined structures.

**Data availability**. Coordinates and structure factors have been deposited in the Protein Data Bank with accession codes 5NJM, 5NLX, 5NM2, 5NM4, 5NQT, 5NM5, 5O5 W and 5NQU. Other data are available from the corresponding author upon reasonable request.

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

## Acknowledgements

We thank SwissFEL for their constant logistic and financial support. We are grateful to Professor Uwe Weierstall for advice on the high-viscosity injector. Dr. Takanori Nakane for his implementation of CHEETAH for EIGER detectors as well as for setting up a tutorial including scripts how to directly process EIGER data. We thank the staff of the X06SA beamline at the Swiss Light Source of the Paul Scherrer Institut for their assistance. Portions of this research were carried out at the Linac Coherent Light Source (LCLS) (LM16 beamtime) at the SLAC National Accelerator Laboratory. LCLS is an Office of Science User Facility operated for the United States. Department of Energy Office of Science by Stanford University. We thank the staff of the CXI beamline at the LCLS for their assistance. We acknowledge the Swiss National Science Foundation for grants 310030_153145 (to G.S.) and 31003A_159558 (to J.S.). P.N. acknowledges support from the European Community's Seventh Framework Program (FP7/2007-2013) under grant agreement no.290605 (PSI-FELLOW/COFUND). G.S. acknowledges support from the ETH Zurich through the NCCR MUST and the ETH FAST programme, as well as support from funding under the H2020 ITN network X-probe under grant agreement no. 637295. We acknowledge support from the Data Analysis Service (142-004) project of the Swiss Universities SUC-P2 program.

## Author contributions

T.W., M.S., M.W. and J.S. conceived the research and designed the experiment. N.O. prepared tubulin samples. R.C., P.N. and M.H. prepared crystals of the A₂ₐ receptor with protein purified by A.D., T.G. and R.M.C. S.B. and U.E. prepared crystals of the MOSTO. L.V. and M.M. prepared lysozyme crystals. T.W., R.C., S.B., P.N., K.J., M.M. and J.S. prepared samples for SMX data collection. D.J., D.G., F.D., E.P., M.W. and T.W. integrated the injector into the beamline setup. D.O. wrote the hit finding program. D.J., R.C., M.L. and V.P. performed SFX data collection at LCLS during a beamtime organized by V.P. and G.S.. T.W., D.J., D.O., D.G., P.N., K.J., S.B., P.S., M.W. and J.S. performed SMX data collection at SLS. T.W., D.O., S.B. and J.S. processed and analyzed the data. T.W., M.S. and J.S. wrote the manuscript with input from all authors.
