## [Peer Review file · Nature Communications]

Reviewers' comments:**Reviewer #1 (Remarks to the Author):**

The authors describe synchrotron structure solution using a high viscosity extruder. The results obtained are impressive with efficient use of both samples and beamtime demonstrated. In general the manuscript is well written and the data presented justify the conclusions drawn. The work does not describe a novel approach but it does demonstrate well the application and development of a relatively new and niche (at least at synchrotron sites) technique into one which might be described as mature.

Some changes to the text and figures could be made to improve presentation and clarity. Specific comments can be found in the attached pdf file.

Reviewer #1 (Remarks to the Author):

Serial millisecond crystallography for routine room-temperature structure determination at synchrotrons
Tobias Weinert et al

Review / specific points

Extrapolation is a wonderful thing but the numbers obtained are an estimate, outside what was done in the experiment. The abstract states that datasets were collected in 3 minutes – this is misleading. Data were collected over 5 hours and extrapolation based on average hit rates etc is used to show a minimal dataset could be collected in 3 minutes assuming several factors are constant throughout the data collection.

It might be noted that three minutes is a long time in modern day MX. Indeed, some of the authors were part of a recent publication (Casanas et al) that demonstrated data collection in < 1 s. While this is currently the exception rather than the rule, data collections of < 10 s are commonplace. I don't mean to undermine the work done here, which is certainly impressive, but there does need to a sense of perspective in this and also in the sample volumes required. Yes, the efficiencies here are high (10 μ L of sample for successful SAD) but this still compares unfavourably with the six crystals needed for 'conventional' structure solution.

The 'latest detector generation' is cited several times. Presumably this differentiates Eiger and Pilatus detectors (this is implied in the discussion)? If so (or if not) this might be made clearer. The advantage over slow detectors such as the 17 Hz rayonix mentioned in the introduction is made clear but not over other pixel detectors such as the Pilatus3. I'm not sure there is anything on page 3/sup table 2 that demonstrates a step-change between what would be obtained with the Pilatus3 and Eiger in experiments such as these. As the sequential nature of hits is not exploited, for 50 (or 100 Hz) data collection the deadtime of the detector would not seem to contribute to the data that can be collected. The case for (future) time resolved experiments is different.

Table S1. What are the number of collected patterns in each case to obtain the 27000 indexed patterns?

I find it difficult to compare the lysozyme datasets in table S1 which is key to part of the decision to use a 20x5 beamsize. In the main text it sounds like the differences are minimal - 'slightly higher over all signal to noise and better cross correlation' but looking at the table the differences appear more significant – the difference is CC1/2 in the outershell is striking. Data with CC1/2 of 0.02... How is the resolution of 1.5Å defined? In the methods section a criterion of CC* of 50% is mentioned but this doesn't seem to be applied here. If this definition is used for all three datasets how do the resolution cutoffs compare? As it stands citing the difference as only an 18% difference in I/sigma seems misleading.

Also here in the main text the authors state that a beamsize of 20x5 was used for the rest of the study, while in the discussion a beamsize of 40x5 is cited for the MOSTO experiments. Please clarify.

The question of resolution cutoffs extends to later in the manuscript where the resolution obtained by different methods is compared. What criteria are used? CC* is mentioned in the methods but this does not seem to have been applied consistently. It is difficult for the reader to really understand if a dataset given to 1.95 with outershell parameters of I/sigma 1.13, CC* 0.79, CC1/2 0.46, Rmeas 91.8 is really inferior to one given to 1.70Å with parameters of 0.44 / 0.45 / 0.11 / 315.4.

A small thing but I find the presentation of the units [ph / (s x um²)] in the first paragraph of the results confusing. It might be clearer and more concise just to give the beamsizes and approximate flux if it is almost the same in each case.

Frame rates of only 50 Hz compared to the maximum of 133 Hz for the Eiger are shown. Is there a reason faster frame rates were not used?

On page 3 a hit rate of 40% is cited. I cannot see any evidence for a number this high. In table 1 a hit rate of > 40% is given for lysozyme but nothing nearly this high for A2A unless I am missing something.

A minor point, but the first reference to figure 6 (page 4) seems misplaced. I don't think the figure shows the map obtained during the beamtime as might be implied by the reader.

Figure 1. I'm not sure the bottom two sections, features and results, add anything for the reader. Certainly the features panel could be removed without any loss of information. Personally I feel a photograph of the setup in situ would be a valuable addition/replacement to at least some of the schematic.

Figure 2. It might be useful for the reader if the number of sites present was given in the legend.

Figure S2 (and discussion). A crystal is typically seen over 5-6 sequential frames. Does the orientation of the crystal change on these or is essentially the same pattern recorded 5 times? Short exposure times to mitigate crystal motions is mentioned in the text. Was this observed when comparing sequential patterns? Was evidence of crystal rotation observed when summing the A2A images (table S2)? Does this partially account for the change in data quality? Whether it is or not would seem to be key to the arguments made in the second paragraph of the discussion.

In the discussion the phrase 'by scanning across the ... with a wide but thin beam' really implies that the beam is being moved. It isn't but this really could be phrased better (as could the moniker 'crystal scanning').

On page 9/top of page 10/later in page 10 supp figure 1 is mistakenly referenced.

Reviewer #2 (Remarks to the Author):

The authors present a set of case studies in which a high-viscosity injector is used in conjunction with an EIGER detector to collect diffraction data from macromolecular crystals embedded in an LCP-matrix at room temperature.

A. Although the results are interesting the proposed setup could potentially be used for other crystal structure determinations, the manuscript has a number of general short-comings that need to be addressed.

1. In the introduction, several lines of related work in the field of serial crystallography approaches with synchrotron radiation are not mentioned at all.

These include:

- the first published work by Gati et al. (2014) doi:10.1107/S2052252513033939
- the work by Nass et al. (2016) <http://dx.doi.org/10.1107/S2052252516002980>
- the methodology published by Coquelle et al. (2015) <http://dx.doi.org/10.1107/S1399004715004514>
- the technique suggested by Schubert et al. (2016). <http://dx.doi.org/10.1107/S2052252516016304>

2. There are a number of unjustified (implicit) generalisations, including:

- in abstract '... allows the majority of crystallographic experiments to be routinely performed at room-temperature'. This is NOT shown in the paper. The four cases used for demonstration are certainly of interest. However, it is not clear at all how mechanically fragile crystals would tolerate (1) pooling of crystals (2) transfer to LCP and (3) the extrusion process.
- in abstract 'Our results show that data collected by room-temperature serial crystallography are of comparable quality to cryo-crystallographic data and can be routinely collected at synchrotrons.' The implied general applicability of the suggested method can NOT be proven by studying four examples with the point in question being addressed only superficially for one of the cases.
- in Discussion 'We demonstrate this approach using microcrystals of the human A2A receptor for which we collected data to 2.14 Å resolution with our SMX method and to 1.95 Å for the same crystals using cryo-crystallography' - a crystal of 30 x 30 x 5 µm³ (Table 1) would nowadays not be considered a micro-crystals anymore.

C. As it stands, the manuscript reads like a sequence of 'advertising statements', while a clear and clean line of arguments leading to the most important conclusions is difficult to detect. It would be good to restructure such that the important arguments can be followed; minor arguments should be put into Supplementary Material, Materials & Methods, figure captions.

D. There are a number of inconsistencies in the text. E.g.

- on page 2 it is stated 'Based on these results we decided to use the 20 x 5 µm² beam for the rest of the study' while then in Table 1 different beam sizes are stated for TD1apo and MOSTO.

E. Nomenclature is sloppy, e.g.:

- 'redundancy' should be replaced by 'multiplicity' throughout
- When the concept of 'correlations' is used it must be stated between which two quantities the respective correlation is calculated.
- The term 'cross correlation' is used rather loosely in different situations.

F. Time-resolution:

- in Discussion. 'Our data collection scheme is very well suited for future time-resolved experiments, because it can combine the ultra-high frame rates of the EIGER detectors (up to 3000 Hz) with laser excitation to achieve time resolutions in the millisecond range and beyond'.

The time-resolution of a time-resolved experiment does not exclusively depend on the frame-rate of the detector. Pump-probe methodology and (chopper-based) shutters have at least an equally strong influence on the achievable time-resolution.

7. SAD phasing

- A2A is not a 'difficult membrane protein target'. The crystals are well diffracting, not exceedingly small, and the sulfur content is normal (ca. 1 S in 30 residues).

8. Materials and Methods / figure legends are sloppily written, e.g.:

- For TD1 crystallization drop volumes are missing.
- Temperature at which LCP mixing was performed is missing.
- MOSTO: 'Crystals were then of a Hamilton syringe' ??
- For the LCLS experiment, photons/pulse is missing
- Radiation doses. 'The radiation damage was calculated' ??
- The dose calculation should take into account that the beam used does not have a top-hat profile in the vertical direction. The 5 μm beam size mentioned probably refers to FWHM of the beam profile - therefore the adjacent regions of crystals will be damaged significantly before the 'central exposure' takes place.
- Figure 3A. What is shown in grey?
- '2FoFc electron density' should be '2Fo-Fc difference electron density', likewise for all other electron densities.

Based on the above points, the manuscript could be reconsidered after substantial revision.

Reviewer #3 (Remarks to the Author):

The manuscript presents results on Serial Microsecond Crystallography using the combination of a high viscosity jet and high frame-rate Eiger PAD to illustrate the possibility of collecting high quality data, sufficient for S-SAD phasing in one case, from challenging real-world crystallographic problems. The manuscript is very well written and comprehensively describes the methods used for sample preparation, data collection and data analysis. The manuscript is well introduced and the discussion and conclusions drawn are sound. There are numerous incremental developments appearing in the literature relating to synchrotron serial crystallography, a number of which are cited in the manuscript. This work constitutes another step forward in this field. In particular, the comparison of SMX, SFX and cryo synchrotron data collection provides a stark illustration of the pros and cons of each method and is very helpful as a reference.

There are two areas of concern that I would like the authors to address or comment on in a revised manuscript:

Firstly, the data collection method introduced in the manuscript uses a jet velocity that moves the sample 5 microns through the beam in order to exactly match the vertical beam size. This ensures full use of any available crystal volume since the horizontal beam size is roughly matched to the width of the jet. Measuring diffraction images from neighbouring 5 micron regions of a crystal may leave the crystal prone to damage from photoelectron or radical species diffusion into unexposed crystal volume. The mean-free-path of photoelectrons will depend on the X-ray energy but some observable effect should be seen at the 5 micron scale [see Cowan, J.A., and Nave, C. (2008). doi:10.1107/S0909049508014623 and Sanishvilli et al. (2011) doi:10.1073/pnas.1017701108]. (It was at this point of reviewing the manuscript that I realised the X-ray energies for the non-6 keV data sets, apart from one, are not explicitly reported). The geometry used may provide some advantage given a horizontally polarised beam but the literature shows the potential for photoelectron damage in the vertical plane even with horizontally polarised light. I would be keen to know if the authors considered this possible damage effect and whether they attempted to

perform any data collections that introduced spacing between the exposed regions for comparison. (REF for this) Even though the authors reported no specific damage it might be that the crystal lattice or B-factors suffered, thereby potentially reducing the diffraction limit. Other radical migration effects may need to be considered also but could be slower. Some discussion in the manuscript on this topic, addressing my questions and comments, is warranted.

Secondly, at the bottom of page 6 the Hadamard Transform method is suggested as a way to deal with very weak data resulting from ultra-high frame-rate time-resolved data sets. There is no signal-to-noise advantage in using the Hadamard method for accessing the weaker individual images and the logic of this statement is therefore not clear to this me. More standard methods for improving weaker data, e.g. appropriate post-refinement and profile fitting would bring larger gains I think – further developments in this area for still-shot FEL and Synchrotron data are still needed and are ongoing in a number of groups worldwide.

Some specific points are:

1. Table 1: X-ray energy for data collection should be added to Table 1 and other Tables in SI.
2. Page 12 – Section entitled “Ensemble refinements of” Line 4. “used Data collected..” needs no capital D.
3. Page 12 – ADP should be defined here.

Manuscript NCOMMS-17-07766-T

Serial millisecond crystallography for routine room-temperature structure determination at synchrotrons

Point-by-point response.

Reviewer #1 (Remarks to the Author):

- *The authors describe synchrotron structure solution using a high viscosity extruder. The results obtained are impressive with efficient use of both samples and beamtime demonstrated. In general, the manuscript is well written and the data presented justify the conclusions drawn. The work does not describe a novel approach but it does demonstrate well the application and development of a relatively new and niche (at least at synchrotron sites) technique into one which might be described as mature. Some changes to the text and figures could be made to improve presentation and clarity. Specific comments can be found in the attached pdf file.*

We thank the reviewer for the positive assessment of our work. The specific suggestions are included in our revised and now clarified manuscript.

Review / specific points

- *Extrapolation is a wonderful thing but the numbers obtained are an estimate, outside what was done in the experiment. The abstract states that datasets were collected in 3 minutes – this is misleading. Data were collected over 5 hours and extrapolation based on average hit rates etc is used to show a minimal dataset could be collected in 3 minutes assuming several factors are constant throughout the data collection.*

We agree with the reviewer that it is generally difficult to arrive to a theoretic time for a data collection after the experiment was done already. In our case the typical data collection can be done in less than one synchrotron shift including injector change, adjustments, test etc. However, it is also clear that in all investigated cases we collected far more data than what was necessary and that we could arrive to a similar result in a much shorter time. To estimate how much is needed we analysed the minimal number of images needed for a converged dataset resulting into the times given in the abstract. We would like to point out that the average we give here is a better representation of the general case, since we could have also chosen data portions with the highest hit rate rather than the average. This would have led to lower, but less realistic numbers as far as the general application is concerned.

To address the point, we have changed the corresponding sentence in the abstract to:

“The datasets for four different target proteins converged after 1000 to 10000 indexed diffraction patterns, which can be collected in 3 to maximally 82 minutes.”

- *It might be noted that three minutes is a long time in modern day MX. Indeed, some of the authors were part of a recent publication (Casanas et al) that demonstrated data collection in < 1s. While this is currently the exception rather than the rule, data collections of < 10s are commonplace. I don't mean to undermine the work done here, which is certainly impressive, but there does need to a sense of perspective in this and also in the sample volumes required. Yes, the efficiencies here are high (10 uL of sample for successful SAD) but this still compares unfavourably with the six crystals needed for 'conventional' structure solution.*

Here we would like to remind the reviewer that it seems not appropriate to compare data collection from a frozen highly diffracting large insulin or lysozyme crystal (as described in Casanas et al) to our method for room temperature structure determination of “real life” samples.

For example, it will be comparatively difficult to solve the A2a receptor by native SAD phasing with the rotation method. Due to the small size of the crystals, especially in the 5 μm dimension one will have to apply a large dose to obtain the resolutions about 2.5 Å that are necessary and the disulfide bridges are sensitive to radiation damage, so one will have to collect data on multiple crystals like demonstrated for a number of targets in Liu et al. 2012 (10.1126/science.1218753). In our case, this would take a very long time. Even the A2a cryo dataset from six crystals included in our manuscript needed several hours full of tedious scanning for diffraction in opaque frozen LCP, crystal alignment and optimization of data collection. In comparison, data collection by our serial method runs fully automatic for many hours once it is setup. This is not only useful for collecting data with high multiplicity and minimal radiation damage as, for example, needed for experimental phasing but also in cases where a high degree of automatization is a plus as for example fragment screening approaches using TD1. Room temperature measurements furthermore are a prerequisite for time-resolved measurements and are less prone to freezing artefacts. In our manuscript, we argue that these advantages more than outweigh longer data collection times.

- *The 'latest detector generation' is cited several times. Presumably this differentiates Eiger and Pilatus detectors (this is implied in the discussion)? If so (or if not) this might be made clearer. The advantage over slow detectors such as the 17 Hz rayonix mentioned in the introduction is made clear but not over other pixel detectors such as the Pilatus3. I'm not sure there is anything on page3/sup table 2 that demonstrates a step-change between what would be obtained with the Pilatus3 and Eiger in experiments such as these. As the sequential nature of hits is not exploited, for 50 (or 100 Hz) data collection the downtime*

of the detector would not seem to contribute to the data that can be collected. The case for (future) time resolved experiments is different.

We thank the reviewer for pointing this out, we should have been more precise concerning the detectors. Indeed, using PILATUS3 detectors with our strategy may be possible. It is good that the referee pointed this out, because it increases the number of beamlines at which the experiments suggested in our manuscript can be performed at. The main difference between PILATUS3 and the EIGER 16M with respect to our experiments is that the EIGER allows continuous data collection with a 99% percent duty cycle, with the dead time (3.8 μ s / internal frame) being evenly spread-out over the requested frame rate. The longer dead times of Pilatus3 (0.95 ms) would lead to a duty cycle of 95% at a frame rate of 50Hz, with the dead time occurring at the end of each frame, leading to a miss of some of the crystal volume in our scanning approach, but this should only marginally affect the SMX method as described here. Certainly, it would compromise the time-resolved experiments we see as the next logical developmental step in the method, as pointed out by the reviewer.

We updated the following sentences to include the suggestion and improve clarity regarding the detectors that can be used:

“The fast readout of the EIGER detector and its duty cycle better than 99%¹⁸ are crucial for this strategy. At the frame rates we used, also PILATUS3 detectors with a duty cycle of about 95% at 50 Hz would appear to be suitable for the strategy presented here, although the higher dead time, which is not evenly spread-out over the individual frames, may lead to a slight decay in data quality.”

“Using 50 Hz frame rates as made possible by high frame rate detectors such as EIGER and PILATUS3, we were able to counteract the negative impact of crystal movement during exposure of each individual frame as realized by using the shutter in our previous work.”

- *Table S1. What are the number of collected patterns in each case to obtain the 27000 indexed patterns?*

For each of the three tested beam sizes we collected a run of 60000 images. With larger beamsizes we observed an increasing number of multiple hits which are more difficult to index. We therefore only used 27000 patterns for each beam size to allow a more accurate assessment of the effect on data quality. All information is now included in Table S1.

- *I find it difficult to compare the lysozyme datasets in table S1 which is key to part of the decision to use a 20x5 beamsize. In the main text it sounds like the differences are minimal - ‘slightly higher overall signal to noise and better cross correlation’ but looking*

at the table the differences appear more significant – the difference is CC1/2 in the outershell is striking. Data with CC1/2 of 0.02... How is the resolution of 1.5Å defined? In the methods section a criterion of CC of 50% is mentioned but this doesn't seem to be applied here. If this definition is used for all three datasets how do the resolution cutoffs compare? As it stands citing the difference as only an 18% difference in I/sigma seems misleading.*

With respect to choosing the right beam it has to be taken into account that the beamsize is also critical for determining the maximal flux. To only compare the effect of the beamsize we therefore reduced the flux of the larger beamsizes to the maximum possible with the 5x5 μm^2 beam. Indeed, this results into the statistics shown for the 5x5 beam in table S1 to be slightly better compared to the larger beams. However, taking the flux reduction into account and the possibility to expose a larger part of the LCP stream, we think the 20x5 beam is the best option in most cases.

The resolution cutoff in Table S1 was defined as a CC* of 0.5 for the medium beam size of 10x5 μm^2 . For an easier comparison data collected with the small 5x5 μm^2 and the large 20x5 μm^2 beam are shown with the same resolution shells. To make this clearer, we have now included an extra line in table S1 with the resolution for every dataset at the commonly used resolution cutoff of CC* 0.5.

We further changed the text in the revised manuscript to:

“However, the effect of the smaller beam was not very pronounced, resulting only in an 18 percent increase in overall I/σ and a 0.09 Å increase in resolution (based on a CC* > 0.5 criterion) as compared to the largest beam tested, despite the flux density being reduced to a quarter in this experiment (see Supplementary Table 1). Based on these results we decided to use the 20 x 5 μm^2 beam for most of the study as crystals from our other three proteins were of comparable size and it allowed us to use the full intensity beam.”

- *Also here in the main text the authors state that a beamsize of 20x5 was used for the rest of the study, while in the discussion a beamsize of 40x5 is cited for the MOSTO experiments. Please clarify.*

The data for the study were collected over the course of four beamtimes. The TD1_{col} data as well as a part of the TD1_{apo} data were collected before our analysis leading to the decision to use a 20 x 5 beam throughout the rest of the study was complete. For MOSTO, we explain in the discussion why the beam was widened (to improve the initially low hit rate). The point of the section on TD1 and MOSTO being rather on data convergence than on data quality, we did not discuss the effects of different beam sizes there.

“...our other three proteins were of comparable size and it allowed us to use the full intensity beam. In some runs, we varied the longer axis of the beam to adjust hit rate and signal to noise depending on the specific characteristics of the sample. The wider beam perpendicular to the flow of the LCP extrusion allowed scanning a larger volume of the crystal-carrying medium. Generally, this led to increased hit rates and thus balanced sample consumption, data collection speed and signal-to-noise ratio.”

Reviewer #2 (Remarks to the Author):

- *The authors present a set of case studies in which a high-viscosity injector is used in conjunction with an EIGER detector to collect diffraction data from macromolecular crystals embedded in an LCP-matrix at room temperature.*

Although the results are interesting the proposed setup could potentially be used for other crystal structure determinations, the manuscript has a number of general shortcomings that need to be addressed.

We thank the referee for the spend time and the suggestions on how to improve our manuscript. All suggestions have been implemented as detailed below.

- *1. In the introduction, several lines of related work in the field of serial crystallography approaches with synchrotron radiation are not mentioned at all.*

These include:

- *the first published work by Gati et al. (2014) doi:10.1107/S2052252513033939*
- *the work by Nass et al. (2016) <http://dx.doi.org/10.1107/S2052252516002980>*
- *the methodology published by Coquelle et al. (2015) <http://dx.doi.org/10.1107/S1399004715004514>*
- *the technique suggested by Schubert et al. (2016). <http://dx.doi.org/10.1107/S2052252516016304>*

Our manuscript did not intent to be a full review which means we wrote the introduction with the intent to guide the reader swiftly towards high viscosity injector based serial crystallography. However, we do understand the concerns of the reviewer and have added a paragraph to point the reader towards alternative techniques that are being used and developed at synchrotrons and are now more or less commonly also called serial crystallography.

“A promising solution is to adapt the methodology developed for macromolecular crystallography at XFELs to synchrotron sources, where radiation damage can’t be outrun but where the radiation dose per crystal can be reduced by using many crystals. Here the term serial crystallography is commonly used in a broader sense, covering loop- or sandwich-scanning techniques¹²⁻¹⁴ as well as the combination of a rotation series with the use of dozens to hundreds of individual crystals^{15,16}. The most direct approach is serial millisecond crystallography (SMX), which utilizes the same high viscosity injectors

successful at XFELs to distribute the radiation dose over thousands of crystals to determine room-temperature structures with minimum radiation-damage”

Beyond our comparison with the A_{2a} structure solved at LCLS we did not want to discuss phasing experiments using XFELs. Hence we initially decided not to include the work by Nass et al. as well as Nakane et al., both demonstrating native SAD phasing at LCLS and SACLA with Thaumatin and Lysozyme, respectively. Nevertheless, now we have included the requested citations in the discussion as follows:

“Based on reports of gadolinium based^{41,42} and native sulfur/chloride SAD phasing at XFEL sources⁴³, phasing exploiting other anomalous scatterers than sulfur should be even easier to implement. “

- *2. There are a number of unjustified (implicit) generalisations, including:
- in abstract ' ... allows the majority of crystallographic experiments to be routinely performed at room-temperature'. This is NOT shown in the paper. The four cases used for demonstration are certainly of interest. However, it is not clear at all how mechanically fragile crystals would tolerate (1) pooling of crystals (2) transfer to LCP and (3) the extrusion process.*

Indeed, we did not want to generalize that the method will work for all samples, but rather that the methods commonly used in crystallography can be routinely performed. To reflect this we changed the abstract, now stating:

“ ...allows most types of crystallographic experiments to be performed.”

The problems the referee describes regarding mechanically sensitive crystals indeed exists, just like in cryo-crystallography, where fishing of mechanically sensitive crystals can be a problem. Additionally, finding the correct cryo-protectant for particularly sensitive crystals is not easy at times. Hence, we added a sentence citing papers demonstrating the use of different high viscosity matrices and also different ways of embedding the crystals.

“Similar to the search for suitable cryo-protectants in classical cryo-crystallography, different crystals may tolerate the embedding process differently. To solve this problem, a variety of different viscous media have been described^{17,20,21}. In our hands, all crystals tolerated LCP very well and hence we only used this well-established medium in the presented study.”

- *- in abstract 'Our results show that data collected by room-temperature serial crystallography are of comparable quality to cryo-crystallographic data and can be routinely collected at synchrotrons.' The implied general applicability of the suggested method can NOT be proven by studying four examples with the point in question being addressed only superficially for one of the cases.*

We agree that it is difficult to deduce general applicability from any case study, which will always be limited in sample size. To address that point and avoid implying that the proposed method will work for any sample just as well, we changed the sentence in question:

“Our results show that data we collected by room-temperature serial crystallography are of comparable quality to cryo-crystallographic data and can be routinely collected at synchrotrons.”

However, for the LCP grown A_{2A}-receptor crystals we do show that the electron density around the ligand binding site is comparable and that the obtained resolution is only about 0.2 Å lower compared to the resolution obtained by cryo-crystallography using the same crystals.

To emphasize the point of comparable data quality a bit more, we also highlighted in the text that our SMX data collected for the soluble TD1_{col}-crystals embedded into LCP extend to only about 0.25 Å lower resolution, despite being collected from much smaller crystals at much lower overall dose per crystal.

“Despite using a much lower dose per crystal, the resolution of our TD1_{col} SMX structure with 2.05 Å is only slightly lower than the 1.8 Å structure of TD1_{apo} obtained by cryo-crystallography from a larger crystal (see Table 1 and Supplementary Table 4). “

- - *in Discussion 'We demonstrate this approach using microcrystals of the human A2A receptor for which we collected data to 2.14 Å resolution with our SMX method and to 1.95 Å for the same crystals using cryo-crystallography' - a crystal of 30 x 30 x 5 μm³ (Table 1) would nowadays not be considered a micro-crystals anymore.*

We disagree on this point, especially due to the small 5 μm dimension that makes data collection extremely dose-dependent leading to difficulties collecting room temperature data on these crystals. In any case, there is no generally accepted definition of a microcrystal but we think most readers would agree our crystals would fall into this category.

- *C. As it stands, the manuscript reads like a sequence of 'advertising statements', while a clear and clean line of arguments leading to the most important conclusions is difficult to detect. It would be good to restructure such that the important arguments can be followed; minor arguments should be put into Supplementary Material, Materials & Methods, figure captions.*

We regret the concerns of the reviewer regarding the structure of our article but would like to point out that this will always depend on the reader and the other two reviewers did not have similar concerns. It would also have been helpful to point out to us which are the points that should be re-structured. Currently we first describe how we optimized data collection parameters in order to allow efficient data collection and data quality. We demonstrate the transfer of a soluble target protein (MOSTO) whose crystals were previously optimized for cryo-crystallography, to show that it can be easy to adapt a system for SMX data collection. Then we show efficient and fast data collection for this and other target proteins using ligand soaking as an all-day example. Then we demonstrate that the data quality is good enough to even allow native SAD phasing in a reasonable amount of time.

Generally, our aim is to show our method to structural biologists and emphasize, based on real-life examples, how mature and readily applicable it already is. Since we aim at the broad readership of Nature Communications, we covered multiple aspects, combining data from a series of samples that may have also been published independently. We feel that splitting up the topics would have made the resulting publications less convincing.

- *D. There are a number of inconsistencies in the text. E.g.*
- on page 2 it is stated 'Based on these results we decided to use the 20 x 5 μm^2 beam for the rest of the study' while then in Table 1 different beam sizes are stated for TD1apo and MOSTO.

Indeed, that was an error, we edited the sentence saying the beam size of 20 x 5 μm was used during most of the study and

“Based on these results we decided to use the 20 x 5 μm^2 beam for most of the study as crystals from our other three proteins were of comparable size and it allowed us to use the full intensity beam. In some runs, we varied the longer axis of the beam to adjust hit rate and signal to noise depending on the specific characteristics of the sample. The wider beam perpendicular to the flow of the LCP extrusion allowed scanning a larger volume of the crystal-carrying medium. Generally, this led to increased hit rates and thus balanced sample consumption, data collection speed and signal-to-noise ratio.”

- *E. Nomenclature is sloppy, e.g.:*
'redundancy' should be replaced by 'multiplicity' throughout

Multiplicity and redundancy are commonly used synonymous throughout crystallographic literature. However, we do agree with the reviewer after looking up the exact definitions, and believe that multiplicity is the more correct term. We replaced it in both occurrences.

- *When the concept of 'correlations' is used it must be stated between which two quantities the respective correlation is calculated. The term 'cross correlation' is used rather loosely in different situations.*

To clarify our discussion we replaced cross correlation in the appropriate places with the more precise CC* including the relevant reference. We use this term since the metric is derived from $\text{CC}_{1/2}$ and we also use it for comparison of our data in other places.

We furthermore edited the following two sentences to improve clarity regarding the ligand SA omit map cross-correlation:

“We generated simulated annealing (SA) ligand omit maps and plotted their cross correlation to the ligand density calculated from the model versus the number of images included in a dataset (See Figure 4).”

“However, ligands already reach 90 % of their final cross correlation to the generated SA omit maps after 4000, 900 and 900 indexed patterns (indicated by dotted lines in the plot)

corresponding to 35, 3 and 5 minutes of data collection for TD1, A_{2A}R SMX and MOSTO, respectively.”

- *F. Time-resolution: in Discussion. 'Our data collection scheme is very well suited for future time-resolved experiments, because it can combine the ultra-high frame rates of the EIGER detectors (up to 3000 Hz) with laser excitation to achieve time resolutions in the millisecond range and beyond'. The time-resolution of a time-resolved experiment does not exclusively depend on the frame-rate of the detector. Pump-probe methodology and (chopper-based) shutters have at least an equally strong influence on the achievable time-resolution.*

We added the following sentence (including references for using choppers or single bunch operation at synchrotrons to achieve high time resolution) in order to point out why we think the detector frame rate is so interesting to use for the collection of time resolved data.

“Other approaches like choppers³⁴, single bunch operation³⁵ or precisely timed and short pulsed lasers are able to increase achievable time resolutions even further, but time resolution obtained simply by detector readout speeds will allow to collect whole time series in a single experiment. “

- *7. SAD phasing: A2A is not a 'difficult membrane protein target'. The crystals are well diffracting, not exceedingly small, and the sulfur content is normal (ca. 1 S in 30 residues).*

We disagree, based on our experience the A2a receptor will indeed be comparatively difficult to solve by native SAD phasing with the rotation method. Due to the small size of the crystals, especially in the 5 μm dimension one will have to apply a large dose to obtain the resolutions about 2.5 Å that are necessary in many native phasing cases. Disulfide bridges are furthermore sensitive to radiation damage, so one will have to collect data on multiple crystals like in Liu et al. 2012 (10.1126/science.1218753), making it a difficult target for this technique.

Nevertheless, to accommodate the concerns we have changed the sentence to:

“Using this strategy, we were able to solve the structure of a human membrane protein pharmaceutical target, the A_{2A} receptor...”

- *8. Materials and Methods / figure legends are sloppily written, e.g.: For TD1 crystallization drop volumes are missing.*

The drop volumes were added.

- *Temperature at which LCP mixing was performed is missing.*

It was added in all cases.

- *MOSTO: 'Crystals were then of a Hamilton syringe' ??*

The sentence has been completed.

- *For the LCLS experiment, photons/pulse is missing*

We added the number of photons/pulse in the methods section, though it was already written in Table one.

- *Radiation doses. 'The radiation damage was calculated' ??*

We changed it to **“The radiation dose was...”**

- *The dose calculation should take into account that the beam used does not have a top-hat profile in the vertical direction. The 5 μm beam size mentioned probably refers to FWHM of the beam profile - therefore the adjacent regions of crystals will be damaged significantly before the 'central exposure' takes place.*

Indeed, the beam shape is FWHM and not top-hat which is being taken into account for the calculation of the diffraction weighted dose per frame in RADDOSE3D. The effect on the consecutive crystal volume should not be too severe since the beam intensity decays with a Gaussian curve. A Gaussian curve with a FWHM of 5 has a σ of 2.12 ($\sigma = \text{FWHM} / 2.35$). The area under such a curve from -2.5 to +2.5 (FWHM) is more than 5 times larger than the area from -7.5 to -2.5 (corresponding to the 5 μm region above the beam that would be fully exposed during the collection of the next frame). Hence, should the beam fully hit the crystal, that dose is less than 20% of the dose received during the full exposure. Given the fact that dose calculations for average crystal sizes with unknown trajectory through the beam can never be more than an estimate, and the dose numbers were already small, we did not take this comparatively small effect into account.

We added a sentence to the description of the dose calculation:

“The dose applied due to the Gaussian beam shape to the region above the beam, that was exposed in each consecutive frame was neglected in the dose calculations, since it is small compared to the dose received in the full exposure.”

- *Figure 3A. What is shown in grey?*

We edited the sentence to explain what is shown in grey (the remaining two $\alpha\beta$ -units completing the heterohexamer).

“The α - and β - subunits present in the asymmetric unit of the SMX structure are shown as red and blue cartoons and the remaining two $\alpha\beta$ -units completing the heterohexamer are shown in grey.”

- *'2FoFc electron density' should be '2Fo-Fc difference electron density', likewise for all other electron densities.*

We thank the referee for discovering this, we changed it throughout.

Reviewer #3 (Remarks to the Author):

- *The manuscript presents results on Serial Microsecond Crystallography using the combination of a high viscosity jet and high frame-rate Eiger PAD to illustrate the possibility of collecting high quality data, sufficient for S-SAD phasing in one case, from challenging real-world crystallographic problems. The manuscript is very well written and comprehensively describes the methods used for sample preparation, data collection and data analysis. The manuscript is well introduced and the discussion and conclusions drawn are sound. There are numerous incremental developments appearing in the literature relating to synchrotron serial crystallography, a number of which are cited in the manuscript. This work constitutes another step forward in this field. In particular, the comparison of SMX, SFX and cryo synchrotron data collection provides a stark illustration of the pros and cons of each method and is very helpful as a reference.*

First, we would also like to thank reviewer 3 for the positive assessment of our work. Certainly, the comparison of the pros and cons of SMX, SFX and cryo synchrotron data collection using the same crystals is an important factor in our publication.

- *There are two areas of concern that I would like the authors to address or comment on in a revised manuscript:*

Firstly, the data collection method introduced in the manuscript uses a jet velocity that moves the sample 5 microns through the beam in order to exactly match the vertical beam size. This ensures full use of any available crystal volume since the horizontal beam size is roughly matched to the width of the jet. Measuring diffraction images from neighbouring 5 micron regions of a crystal may leave the crystal prone to damage from photoelectron or radical species diffusion into unexposed crystal volume.

- *The mean-free-path of photoelectrons will depend on the X-ray energy but some observable effect should be seen at the 5 micron scale [see Cowan, J.A., and Nave, C. (2008). doi:10.1107/S0909049508014623 and Sanishvilli et al. (2011) doi:10.1073/pnas.1017701108]. (It was at this point of reviewing the manuscript that I realised the X-ray energies for the non-6 keV data sets, apart from one, are not explicitly reported). The geometry used may provide some advantage given a horizontally polarised beam but the literature shows the potential for photoelectron damage in the vertical plane even with horizontally polarised light. I would be keen to know if the authors considered this possible damage effect and whether they attempted to perform any data collections that introduced spacing between the exposed regions for comparison. (REF for this) Even though the authors reported no specific damage it might be that the crystal lattice or B-factors suffered, thereby potentially reducing the diffraction limit. Other radical migration effects may need to be considered also but could be slower. Some discussion in the manuscript on this topic, addressing my questions and comments, is warranted.*

We agree that this type of radiation damage may occur. It has to be noted however, that the dose per crystal is not larger than what is generally accepted for room temperature

structures obtained with the rotation method, which leaves a lot more time for photoelectrons and radicals to cause their effects on all regions of the crystal. In addition, the refined overall B-factor of our SFX A2a structure is very similar to the refined B-factor of the SMX structure. Although this does not prove that our method is as damage free as SFX it is an indication that the extent of global radiation damage is not severe.

With our experimental setup, it would be difficult to analyse these damage effects, since the crystal translates during exposure and hence we cannot compare relative diffraction intensity loss as a single frame radiation damage indicator like used in Sanishvilli et al. (2011). Shutting the beam during exposures to analyse radiation damage is a good suggestion. We did not try such a data collection pattern so far. However, due to the lack of a suitable one pattern metric for the extent of radiation damage we would be limited to the electron density as an indicator of diffraction quality and the electron density already does not show damage effects when data were collected without spacing in the exposed regions and hence the effects might be difficult to judge.

- *Secondly, at the bottom of page 6 the Hadamard Transform method is suggested as a way to deal with very weak data resulting from ultra-high frame-rate time-resolved data sets. There is no signal-to-noise advantage in using the Hadamard method for accessing the weaker individual images and the logic of this statement is therefore not clear to this me.*

The Hadamard transform cannot improve signal to noise ratios, but by using multiple merged images and leaving out only one for each recorded time point while calculating differences based on the Hadamard transform may allow indexing and integration to work well even at very high frame rates, which would result in extremely sparse data that will not be processable on a per-frame basis. We edited the sentence to point that out more clearly:

“To allow processing of high frame-rate data, several images can be merged together, resulting in lower frame rates but stronger diffraction patterns to allow indexing, integration and background subtraction. Leaving out individual images from the merged images and using the Hadamard transform³⁶ to deconvolute them, may allow time resolved analysis far beyond indexable individual diffraction patterns at these frame rates.”

- *More standard methods for improving weaker data, e.g. appropriate post-refinement and profile fitting would bring larger gains I think – further developments in this area for still-shot FEL and Synchrotron data are still needed and are ongoing in a number of groups worldwide.*

The reviewer is right, there is fast progress in the processing of serial crystallographic data and these ongoing efforts will only improve the applicability of our method. To point this out more clearly, we added the sentence below and appropriate references to the revised version of our manuscript:

“Since the advent of single shot serial crystallography data processing progressed significantly by improving detector geometries³⁷ as well as including prediction- and post-

refinement³⁸⁻⁴⁰ we expect these developments to continue at their fast pace, allowing challenging time resolved experiments to be even more feasible.”

▪ *Some specific points are:*

1. Table 1: X-ray energy for data collection should be added to Table 1 and other Tables in SI.

2. Page 12 – Section entitled “Ensemble refinements of ….” Line 4. “used Data collected..” needs no capital D.

3. Page 12 – ADP should be defined here.

We thank the reviewer for pointing out these specific points, we have included the changes into the manuscript.

REVIEWERS' COMMENTS:

Reviewer #1 (Remarks to the Author):

In general I feel the authors have addressed all the points raised and the manuscript is now suitable for publication.

I did notice a clear mistake in table S1 for lysozyme 20x5 however which should be resolved first.
Resolution at which CC* drops to 0.5 given as 1.59A
Resolution of dataset is given as 1.58A with CC* = 0.19

If CC* drops by this much in 0.01A there are issues with the data.

Either the outershell statistics are given incorrectly or the resolution at which CC* = 0.5 is given incorrectly. Looking at the other statistics (CC1/2) it seems likely that the resolution at which CC* = 0.5 is given incorrectly. This seems to be a genuine mistake but it needs to be addressed. When it is the text on page 5 of the main text also needs to be updated.

Reviewer #3 (Remarks to the Author):

I am satisfied that the authors have address, or at the very least responded, to all the concerns raised and have made all required corrections.

Reviewer #1 (Remarks to the Author):

In general I feel the authors have addressed all the points raised and the manuscript is now suitable for publication.

I did notice a clear mistake in table S1 for lysozyme 20x5 however which should be resolved first.

Resolution at which CC^* drops to 0.5 given as 1.59A

Resolution of dataset is given as 1.58A with $CC^* = 0.19$

If CC^* drops by this much in 0.01A there are issues with the data.

Either the outershell statistics are given incorrectly or the resolution at which $CC^* = 0.5$ is given incorrectly. Looking at the other statistics ($CC_{1/2}$) it seems likely that the resolution at which $CC^* = 0.5$ is given incorrectly. This is seems, to be a genuine mistake but it needs to be addressed. When it is the text on page 5 of the main text also needs to be updated.

We checked our data carefully and this effect is due to an option used in partialator, that applies a per-pattern resolution cut-off, leading to a very sharp drop in the amount of reflections and the completeness at this boundary. In the table in question we give the statistics of the 5 x 5 um and the 20 x 5 data set to the same resolution range as for the 10 x 5 um data set (whose resolution cutoff was chosen based on CC^* drop below 50%) in order to allow for comparison. Therefore, the statistics for 5 x 5 and 20 x 5 in this table don't follow our cut-off criteria used in the rest of the manuscript.

We added a star to the value in supplementary Table 1 with the explanation for the sharp drop of CC^* :

“* the sharp drop in CC^* is due to the lack of data in the shell, resulting in the use of the “pushres” option in partialator, that applies a per-pattern resolution cutoff based on observed peaks.”